# Graded phononic metamaterials based on scalable microfabrication and design

Charles Dorn [1,2,5], Vignesh Kannan [1,3,5], Ute Drechsler[4] & Dennis M. Kochmann [1] ✉

Metamaterials' engineered internal structures enable customized material properties beyond those found in nature, such as the capability to guide, attenuate, and focus waves at will. Phononic metamaterials aim to manipulate mechanical waves, with broad applications in acoustics, elastodynamics and structural vibrations. A key bottleneck in the advancement of phononic metamaterials is their scalability beyond tens of unit cells per spatial dimension, which equally affects their design, simulation, and fabrication. Here, we present a framework for scalable inverse design of spatially graded phononic metamaterials for elastic wave guiding, together with a scalable microfabrication method. This framework enables the design and realization of complex waveguides including hundreds of thousands of unit cells, potentially extendable to millions with no change in protocol. Scalable designs are optimized with a ray tracing model for waves in spatially graded beam lattices and fabricated by photolithography and etching of silicon wafers, to create free-standing microarchitected films. Wave guiding is demonstrated experimentally by using pulsed laser excitation and interferometric displacement measurements. Broadband wave guiding is demonstrated, indicating the promise of our scalable design and fabrication methods for on-chip elastic wave manipulation.

The nature of wave propagation in periodic lattices drives the fundamental behavior of materials, from the electronic band structure to thermal and optical properties. Phononic metamaterials build a bridge from wave phenomena at the atomic[1] to engineering scales, where lattices can be carefully designed to manipulate elastic waves[2]. As rapidly advancing manufacturing technology has enabled the fabrication of intricate geometries, a rich literature on phononic metamaterials has emerged in pursuit of harnessing waves for broad applications, including vibration isolation[3,4], sensing[5], and energy harvesting[6].

A key limitation that plagues the advancement of phononic metamaterials is the scalability of both computation and fabrication[7–9]. Modeling, design, and manufacturing efficiency remain bottlenecks for architectures spanning tens of thousands to millions of unit cells, which aim to dissolve the differences between materials and structures. Solving the problem of scalability promises an enlarged and widely untapped design space beyond periodic architectures. Extension of the design space to non-periodic architectures introduces wave manipulation functionalities out of reach of periodic architectures. Examples include wave guiding along topological interfaces between different unit cell architectures within a bandgap[10] and the introduction of defects for localizing frequencies within a bandgap[11]. Alternatively, spatial grading (introducing smooth variations of the unit cell in space) enables wave guiding and focusing within a pass-band[12] with potential for broadband applicability. Scaling spatial grading to architectures with many unit cells opens a large design space of

[1]Mechanics and Materials Laboratory, ETH Zurich, Zurich, Switzerland. [2]Department of Aeronautics and Astronautics, University of Washington, Seattle, WA, USA. [3]Laboratoire de Mécanique des Solides, CNRS, École Polytechnique, Institut Polytechnique de Paris, Palaiseau, France. [4]IBM Research—Zurich, Ruschlikon, Switzerland. [5]These authors contributed equally: Charles Dorn, Vignesh Kannan. ✉e-mail: dmk@ethz.ch

complicated multiscale spatial grading profiles. This expansion of the design space can enhance applications that spatial graded phononic metamaterials already show promise for (such as energy harvesting[13] and signal processing[14]), as well as uncover new wave manipulation capabilities.

On the computational side, the efficiency of modeling and design methods is a bottleneck to scalability due to the multiscale nature of phononic metamaterials. While efficient multiscale modeling and design methods are available for periodic architectures thanks to Bloch's theorem[1], periodic architectures barely scratch the surface of the vast metamaterial design space. Looking beyond periodic architectures, spatial grading promises enhanced functionalities such as wave focusing[15], broadband attenuation[16,17], and signal processing[18]. However, graded architectures are more complicated to model than periodic architectures, since multiple length scales must be resolved, compared to modeling a single unit cell of a periodic architecture. High-fidelity computations (such as transient dynamic finite element analysis) scale poorly to large architectures, where each unit cell must be modeled, which becomes increasingly expensive for inverse problems that require many evaluations of a forward model. Therefore, most existing spatial grading design methods have relied on heavily restricted design spaces (for example, linear gradings[19] and analytical solutions[20]) or restricting assumptions of long wavelengths to enable homogenization[21,22]. Specialized homogenization methods have been developed to capture dispersive effects at finite frequency[23,24]. High-frequency homogenization offers means of approximating dispersion relations, which have been used, for example, to design metastructures consisting of multiple unit cell domains to achieve localized energy[25] (rather than modeling spatial gradings). Thus, there is a rich but largely unexplored design space of spatially graded phononic metamaterials spanning large numbers of unit cells.

The authors' previous work[26,27] offers a foundation for the design of complex spatial gradings using ray theory (which is not restricted to the low-frequency homogenization regime and is based on the full local dispersion relations in the neighborhood of each unit cell). Ray tracing provides a computationally efficient tool for modeling wave motion through graded architectures that enables optimal design (see Supplementary Information Section 2.3 for a discussion of computational efficiency). In this work, our design methodology generalizes our previous formulations[27] to capture physically realistic beam-based architectures and broader design objectives that enable scalability through modular design.

On the experimental side, the fabrication of 2D wave-guiding phononic metamaterials using standard machining and 3D-printing techniques is limited to tens to hundreds of unit cells[19,28,29], making it difficult to truly fabricate metamaterials rather than structures. Going from tens to hundreds of thousands of unit cells at the meso-scale requires the adaptation of specialized microfabrication techniques that have rarely been exploited for free-standing elastic wave guides[30]—especially spanning tens of micrometers to millimeters. The current state of the art in commercial macroscale 3D printing technology enables architectures with hundreds of unit cells per dimension[31]. These methods, which fall within the broad category of additive manufacturing (AM), have been well understood for polymeric materials that, unfortunately, are not suitable for wave propagation because of material damping. Although exciting new directions in metal and ceramic AM have emerged[32–34], there is still much work to be done to control printing resolutions and material properties. Microfabrication techniques enabled by multi-photon lithography, traditionally used with polymeric materials at much superior spatial resolution[35], have breached this limitation[36,37]. Such techniques, while growing rapidly, are still nascent in their ability to achieve large sample sizes close to 100 mm while maintaining sub-micrometer spatial resolution[38,39], lesser so when spatially-graded non-periodic architectures are involved. Scalable fabrication of low-damping 2D elastic waveguides is especially challenging for nano- and microscale manufacturing, where two-photon lithography[35,40–44] and its variants[45] can hardly scale beyond samples with tens to hundreds of unit cells per dimension in a tractable build time. The current state of the art in fast fabrication rates is still limited to polymeric materials[46]. Microfabrication of optical metamaterials has achieved better scalability, with two-dimensional architectures reaching over a billion unit cells, using electron beam lithography and atomic layer deposition[47,48]. Such applications traditionally require smaller sample sizes than those required here to guide elastic waves. At such small length and time scales, the excitation and measurement of waves becomes a challenge, which often requires custom-developed pump-probe optical systems[49–51].

In this work, we introduce a solution to phononic metamaterial scalability both computationally and experimentally. Our results enable the design and realization of elaborate spatially graded metamaterials spanning at least three orders of magnitude in length scales with tens of thousands of unit cells. To achieve scalable computational design, we leverage an optimization framework based on ray tracing for efficient modeling of wave propagation. A modular design approach is developed, where multiple tiles are independently designed to achieve different wave guiding functionalities. The tiles act as building blocks that are assembled like puzzle pieces into large designs with customizable wave guiding capabilities. To realize our designs, microfabrication based on photolithography and etching of silicon wafers creates free-standing architected films. Experimental demonstration of wave guiding is achieved using a nanosecond pulsed laser for excitation and heterodyne interferometry for measurement of particle displacements. The demonstrated design and microfabrication concept open doors to new applications, such as on-chip vibration isolation and signal processing for microscale electromechanical systems (MEMS).

## Results
### Modular waveguide design
Spatially grading the properties of a material is a powerful tool for manipulating propagating waves, which has long been used, for example, for gradient-index optics[52]. Extending this idea to grading the unit cells of phononic metamaterials introduces a multiscale problem, which is computationally challenging to simulate and optimize. Brute-force transient simulations (for example, using finite elements) are accurate but prohibitively slow for inverse design of large microstructures. Hence, existing work has relied primarily on simple, intuitive designs (such as radially symmetric and linear gradings[12,15,19,53]), leaving a rich untapped design space of complex spatial gradings to be explored.

To circumvent the computational efficiency bottleneck, we leverage ray tracing to model wave motion in graded phononic metamaterials, generalizing well-established ray theories for smooth continua (for example, seismic ray theory[54] and geometric optics[55]) to graded metamaterials[26]. Ray tracing in phononic metamaterials provides an efficient modeling tool, which relies on local Bloch wave analysis in the neighborhood of each unit cell to compute local dispersion relations (assuming smooth spatial gradings and hence locally an approximately periodic medium). Consequently, this method applies both within and above the low-frequency homogenization limit, since the complete dispersion relations are accounted for. The resulting spatially variant local dispersion relations act as a Hamiltonian for tracing rays to determine how waves propagate in graded phononic metamaterials. While ray tracing is appealing due to its efficiency, it is an approximation under the assumption that spatial grading is slow relative to relevant wavelengths, which inherently introduces some error. Rigorous theoretical error analysis given complicated grading profiles and general dispersion relations is an open problem; here we ensure that ray tracing solutions are representative by comparison to transient finite element simulations.

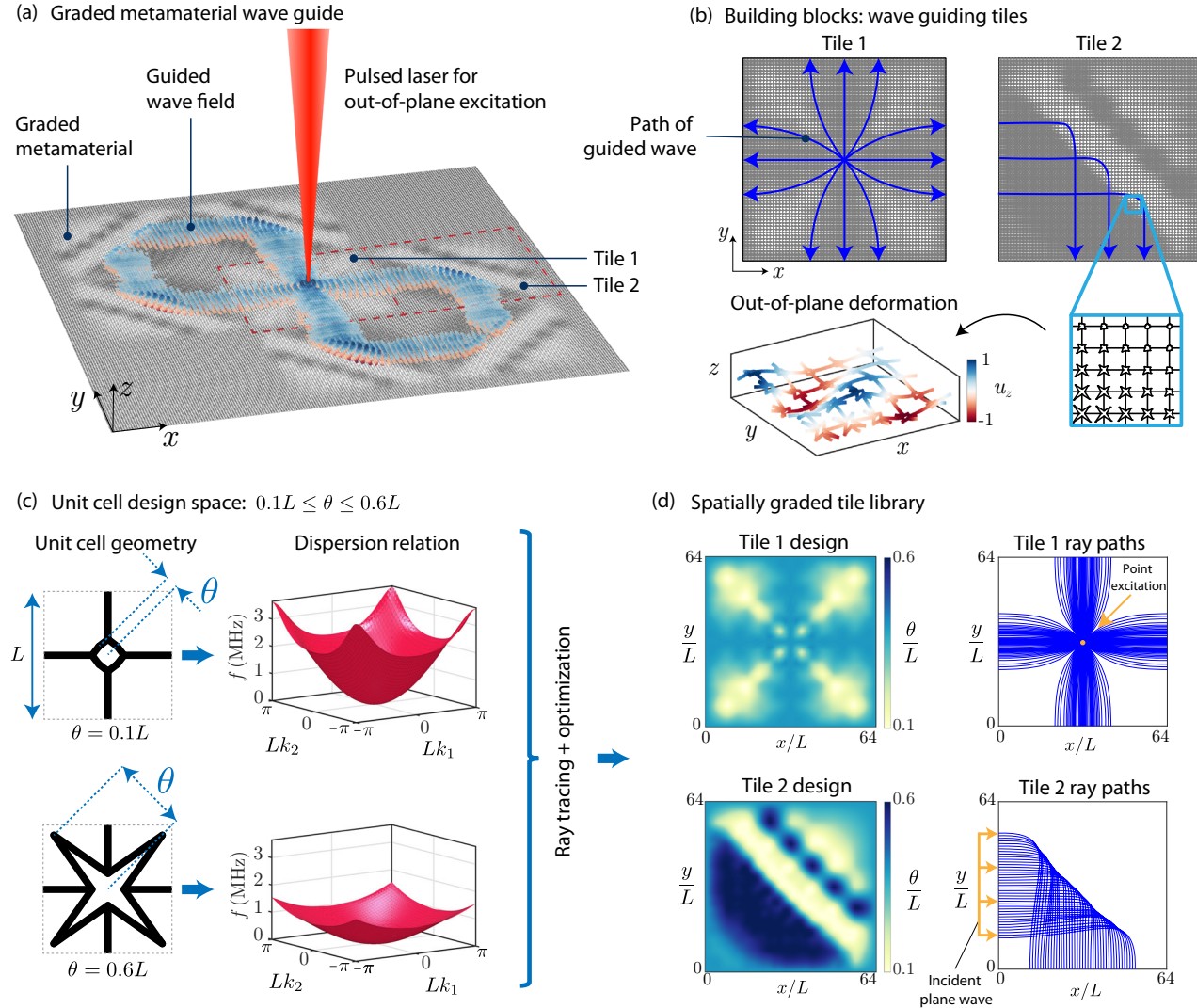

**Fig. 1 | Spatially-graded phononic metamaterial design concept. a** Schematic illustration of the metamaterial composed of tiles, whose wave properties are carefully designed through the spatially varying unit cell geometry. Specifically, waves in the out-of-plane displacement $u_z$ are considered. **b** The design is built by assembling tiles, each of which has spatial grading designed for a specific wave manipulation functionality. **c** The unit cell geometry is modulated by varying the unit cell design parameter $\theta$ (shown are two example configurations and the corresponding wave dispersion relations). **d** Library of tile designs and the corresponding wave motion illustrated by ray trajectories: Tile 1 splits the wave emanating from a point excitation, while Tile 2 redirects a plane wave by 90°.

In this work, we adopt ray tracing for efficient forward modeling, which is crucial to achieve scalability to large and complicated graded geometries. An optimization-based inverse design framework is built around ray tracing by extending our previous formulations[27]. Specifically, an optimization problem is posed to design the spatial distribution of unit cells such that the corresponding ray paths are shaped in a prescribed way. Consider the unit cells in Fig. 1c, in which each edge represents an elastic beam. The unit cell has one design parameter $\theta$, which is restricted to the range $0.1L \leq \theta \leq 0.6L$ for manufacturability ($L$ is the height and width of the square unit cell). The foundation for forward modeling using ray tracing is the local dispersion relation throughout the metamaterial, which is computed using a beam finite element model[28,56]. Focusing on out-of-plane bending vibrations, as in our experiments, the lowest dispersion surface is plotted in Fig. 1c for the two extremes of the design space. For this unit cell, the lowest out-of-plane dispersion surface does not intersect other out-of-plane dispersion surfaces, so it can be studied in isolation. We aim to optimally design the spatial distribution of $\theta$ in a metamaterial consisting of many unit cells to achieve a prescribed wave guiding objective through the resulting spatial variation of the (local) dispersion relations.

An optimization problem is solved to shape the ray trajectories in two spatially graded square tiles, shown in Fig. 1d, each spanning $64 \times 64$ unit cells. The first design, Tile 1, considers a point excitation at the center unit cell (from which the rays originate) and designs the spatial distribution of $\theta$ to guide the resulting wave to exit the tile along the four tile edges with rays perpendicular to the edges. The second tile, Tile 2, considers an incident plane wave with horizontal rays entering from the left tile boundary and redirects the rays to exit perpendicular to the bottom tile edge, thus rotating the incident plane wave by 90°. Both tiles are designed to have $\theta = 0.4L$ around the perimeter, to ensure compatibility between tiles when assembled into the metamaterial (Fig. 1a). The distribution of $\theta$ and the corresponding ray trajectories are plotted in Fig. 1d for both tiles. Details of the optimization setup for tile design are presented in the Supplementary Information Section 1. This optimization setup is generally formulated to handle a wide range of objectives, so additional tiles achieving functionality beyond that of Tiles 1 and 2 could be developed in future work.

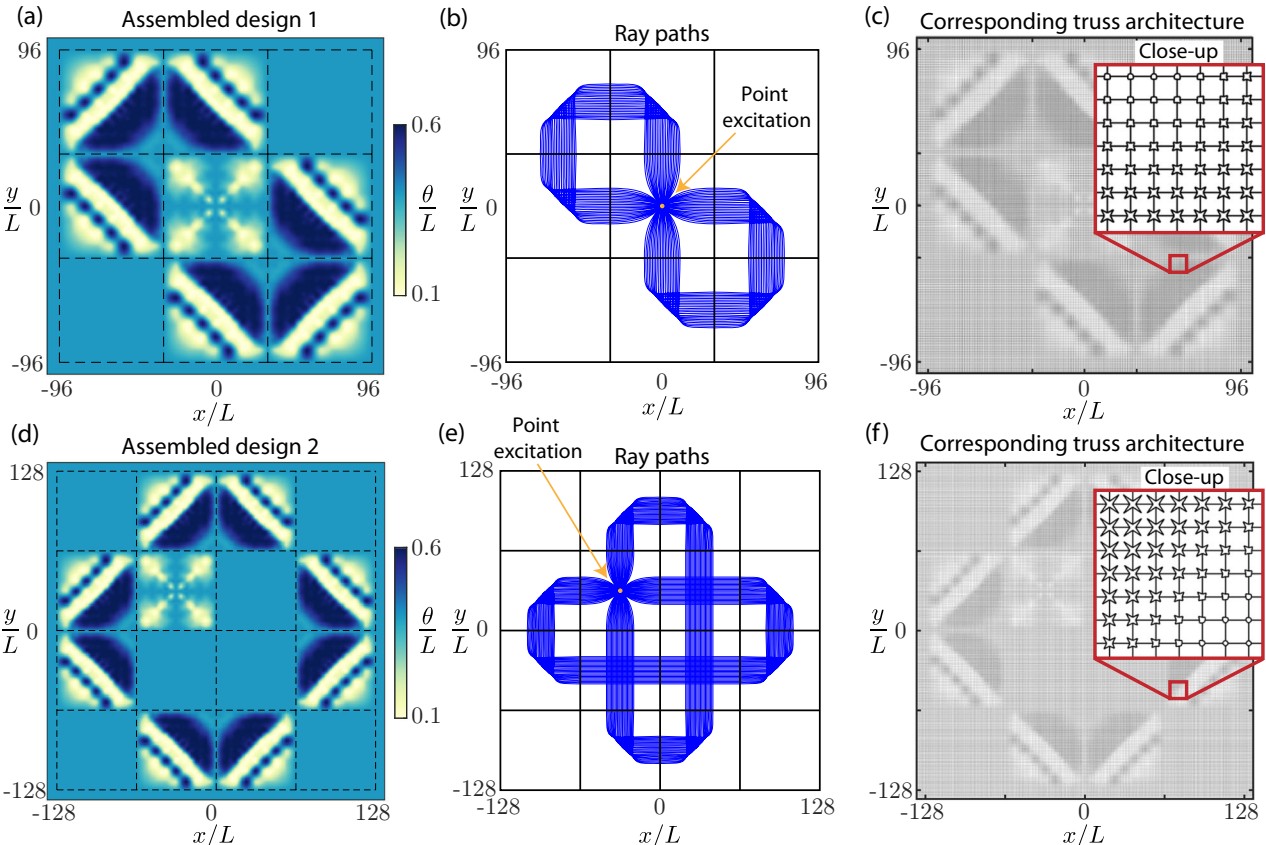

**Fig. 2 | Modular design by tile assembly. a** Spatial distribution of design parameter $\theta$ for an assembly of 3 × 3 tiles. **b** Ray trajectories for the assembly in (**a**) in response to a point excitation at the center. **c** Graded phononic metamaterial containing 192 × 192 unit cells, realizing the design distribution in (**a**) to result in the figure-eight wave motion in (**b**). **d** Spatial distribution of design parameter $\theta$ for an assembly of 4 × 4 tiles. **e** Ray trajectories for the assembly in (**d**) in response to a point excitation. **f** Graded phononic metamaterial containing 256 × 256 unit cells, realizing the design distribution in (**d**) to result in the complex wave motion in (**e**).

Since the two tiles are designed to be compatible, there is no jump in $\theta$ at the shared boundary when the two tiles are placed next to each other. Consequently, the ray paths of Fig. 1d can be continuously connected between adjacent tiles. This allows for solutions to be assembled by placing tiles next to each other such that the rays of each tile are connected to form a desired set of ray paths. Figure 2a provides an example of a tile assembly, which plots the spatial distribution of $\theta$ with dotted lines highlighting the boundary between tiles. The corresponding ray trajectories are shown in Fig. 2b, which are connected to form a figure-eight shape. Thus, waves emerging from out-of-plane excitation at the origin are guided along the figure-eight. The geometric realization of this assembled design is shown in Fig. 2c, where the geometry of each unit cell is determined from the designed distribution of $\theta$ in Fig. 2a. Note that, since all rays in the figure-eight return to the point of excitation, the design – in the ideal case without dissipative losses – would result in continued traversal of the figure-eight even after the point excitation is removed.

Design by tile assembly is a scalable approach that leads to many possible designs. While brute-force optimization of ray trajectories on a large domain could be formulated, such an approach scales poorly with increasing domain size compared to modular design by tile assembly. Once each tile is designed, connecting them to achieve large-scale designs is trivial and introduces no additional computational cost. A second example of assembled tiles is shown in Fig. 2d. Its ray trajectories guide the wave emerging from a point excitation along the outline of a cross, as shown in Fig. 2e. The corresponding beam architecture is shown in Fig. 2f.

Transient finite element simulations are performed to validate the designs. Supplementary Videos 1 and 2 show animations of the finite element simulation results for the case of harmonic excitation at the design frequency. The transient finite element simulations take on the order of 5 hours, highlighting the necessity of ray tracing for inverse design, as it takes on the order of seconds to trace the rays of a tile. Supplementary Information Section 2 provides details about the finite element model setup and analysis of computational efficiency. Additionally, we note that ray trajectory design is performed for a specific target frequency but—in both simulations and experiments— wave guiding is observed over a broad frequency band surrounding this target frequency (see the Supplementary Information Sections 2 and 5 for details).

This modular approach offers a means of designing broadband phononic metamaterial waveguides spanning a large number of unit cells, taking full advantage of spatial grading. The examples of Fig. 2a, d span approximately 37,000 and 66,000 unit cells, respectively, and this approach directly scales to larger designs simply by assembling more tiles (see Supplementary Information Fig. 2). Thus, through efficient modeling and optimization based on ray tracing together with a modular tile assembly approach, we can circumvent the challenges of computational scalability to compute large and elaborate metamaterial designs. The challenge that remains is scalable fabrication to realize such designs, for which we turn to semiconductor microfabrication, drawing inspiration from chip manufacturing methods.

## Microfabrication: Silicon wafers to beam-based phononic metamaterials

Fabrication of spatially graded architectures spanning multiple length scales, such as those in Fig. 2, is out of reach of standard manufacturing

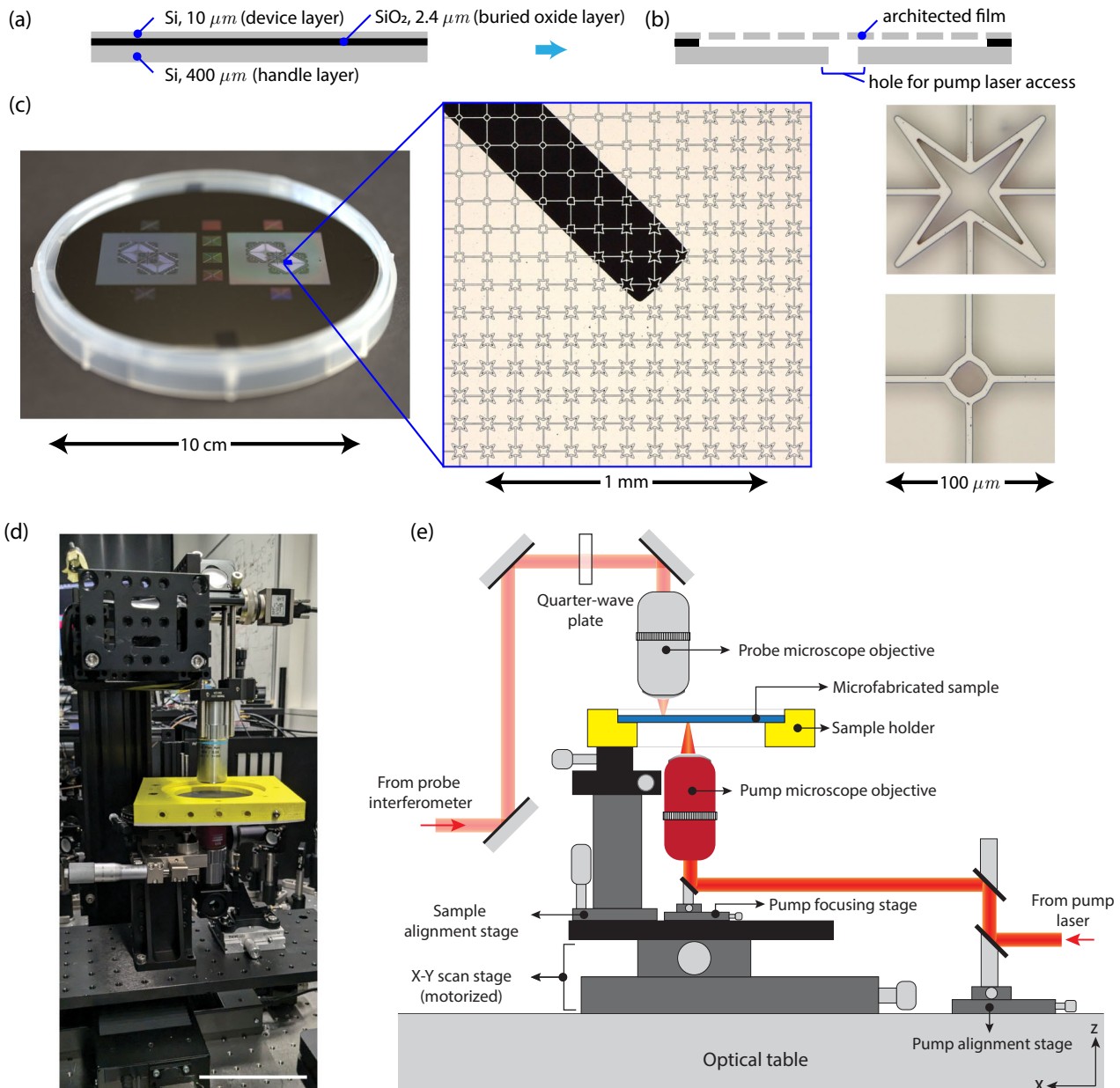

**Fig. 3 | Fabrication and experimental setup.** Schematic of the silicon-on-insulator wafer sample (**a**) before and (**b**) after microfabrication (layer thickness is not drawn to scale). **c** Micrographs of a prototype, showing the full wafer, a magnified top view of 13 × 13 unit cells (with a bottom hole visible in black), and two example unit cells corresponding to the two design space extremes of Fig. 1c. (**d**) Photograph and (**e**) schematic of the experimental setup used to excite and probe wave motion in wafer-based graded metamaterials. The scale bar in (**d**) corresponds to 100 mm.

methods at the macroscale. We adopt silicon microfabrication techniques for a solution.

A schematic of the wafer cross-section after fabrication is shown in Fig. 3b and micrographs of the prototype in Fig. 3c. Two samples are fabricated on a single wafer to reduce the number of wafers used. The figure-eight design prototype spans a 3 × 3 cm region with a square unit cell of $L = 100\,\mu m$ side length. Individual beams are $5\,\mu m$ wide. To enable excitation via a pulsed laser in the experiments, holes are etched in the substrate layer to allow the pulsed laser to excite the device layer from the backside.

In practice, the entire wafer can be architected with micron-scale features, with the only scalability limitation being the size of the wafer itself. We have demonstrated this scalability by also manufacturing ~ 600, 000 unit cells in a single SOI wafer of 100 mm diameter in a separate study[57]. We thus exploit the state of the art in microfabrication technology to manufacture scalable phononic waveguides with micrometer feature resolution of sizes up to 80 mm in diameter. Microfabrication technology has pushed resolution limits down to the nanometer scale[35,58–61], including advances from the fields of micro- and nano-electromechanical devices (MEMS/NEMS)[30,62–66]. Most of those studies focused on devices bonded to a substrate or free-standing nano-architectures no larger than hundreds of micrometers. Using the aforementioned well-established techniques, we target a mesoscale, where the functional unit cells span micrometers, and the free-standing structure spans almost a full silicon wafer. At these scales, our computational-fabrication-experimental framework achieves robust control of spatially graded architecture. This admits controlling wave motion in the tens of kHz to MHz regime across centimeters—a regime of long-standing interest to the field[16,19,28,29,67].

### Experimental wave guiding demonstration

Experiments were performed to demonstrate wave guiding in the prototype, using a pump-probe setup. A 1030 nm nanosecond pulsed laser was used for photoacoustic pump excitation, sending broadband out-of-plane elastic wave modes through the sample. The aluminum film on the sample surface enables the transduction of optical to mechanical pulses for wave excitation. The out-of-plane displacement response was measured by a custom-built heterodyne interferometer. A photograph and schematic of the experimental setup are shown in Fig. 3d, e. Due to the repeatability of elastic wave propagation, data is collected by successively pumping the excitation pulse at the same spatial location, while the probe laser scans different positions on the wafer. The signal-to-noise ratio was excellent at frequencies up to 2 MHz and measurable up to 4 MHz. This was possible due to the low intrinsic damping of single-crystal silicon and the high displacement resolution of the interferometer. Though not necessary here, any dissipation arising from the surrounding air can be mitigated by experiments in a vacuum. The experiment, hence, has a unique potential to probe wave attenuation due to architecture alone.

To capture the metamaterial sample's ability to guide elastic waves, data is collected along three lines, at $x = 0$ (denoted L1), $x = -32L$ (L2), and $x = -64L$ (L3), see Fig. 4a. At 100 measurement points along each line, the time series displacement signal is measured immediately following a pump excitation. Pump excitation occurs at the same point on the sample for all measurement scans. Figure 4c shows the measured response at each position along L1, L2, and L3, comparing the experimental results to simulated data from a finite element model (see Supplementary Information Section 2). The signal-to-noise ratio was excellent for all measurement points except two (marked by the black arrow for line scan L2). For reference, Fig. 4a shows the maximum displacement at each spatial location during a transient finite element simulation, indicating each of the line scans with respect to the guided wave path.

The measured response of the wafer clearly captures wave guiding along the designed figure-eight trajectory. The experimental data closely match the finite element simulation data, with agreement in both the location and timing of large displacement amplitudes. Furthermore, while the waveguide is designed for a specific target frequency of 750 kHz, broadband waveguiding is observed. A detailed frequency analysis is presented in Supplementary Information Section 5, showing that wave guiding is achieved in the window of 250-800 kHz, both in experimental and simulated data. The experiment further shows a different wave guiding response above 800 kHz, which has been validated using finite element analysis (see Fig. 4b, d). The latter was not explicitly introduced during inverse design. Thus, the proposed methodology is capable of customized manipulation and discovery of phenomena across a frequency range broader than that of the inverse design. The observed broadband wave guiding likely stems from having dispersion relations with approximately self-similar isofrequency contours and angular distributions of group velocity (which drive ray trajectories) over the frequency range 250-800 kHz for all unit cells in the chosen design space.

## Discussion

This work pushes the limits of phononic metamaterial scalability in computational design and experiments. We have presented an inverse design framework together with a microscale wafer fabrication method, both of which are demonstrated on designs spanning hundreds of unit cells per dimension with potential for further scalability. Our design method relies on optimization of an efficient ray tracing model of wave motion in spatially graded phononic metamaterials. A versatile modular approach involves designing individual tiles that are assembled to achieve complex wave-guiding objectives. This leads to designs spanning multiple length scales. To realize those designs, we adapt wafer manufacturing methods to create free-standing architected films, with the scalability to fill an entire wafer with millions of unit cells. Experiments based on a pump-probe scheme for excitation and measurement demonstrate broadband wave guiding capabilities of an optimized design.

Unlocking the scalability of phononic metamaterials promises both scientific and technological advances. On the scientific side, the presented methods enable a true separation of length scales to access the material rather than the structural regime, providing a valuable setting for future experimentation. Furthermore, the proposed microfabrication technique paves the way for high-throughput experimentation by the fabrication of many samples on a single wafer. On the technological side, the presented scalable approaches greatly expand the design space of phononic metamaterials, consequently expanding their functionality for applications from vibration isolation in MEMS[68] to high-frequency energy harvesting[69,70] to microfluidics[71], also leveraging the demonstrated broadband stability of the waveguide. The presented waveguide designs serve as illustrative examples, in which wave guiding along customized trajectories is enabled by scalability to large designs.

## Methods

### Microfabrication

We use cleanroom-based microfabrication processes on commercially procured 100 mm Silicon-On-Insulator (SOI) wafers to manufacture the figure-eight design prototype of Fig. 2a–c. The graded structure of Fig. 2c is fabricated in the top (device) layer of an SOI wafer. A schematic of the cross-section of an as-received SOI wafer is shown in Fig. 3a. Our fabrication process involves photolithography and deep reactive ion etching to architect the device layer, followed by removal of the buried oxide (SiO$_2$) layer, using vapor hydrofluoric acid etching. This results in a free-standing architected film supported by the substrate only around the outer perimeter, similar to a drum with an architected membrane. Additionally, windows are etched in the supporting substrate to enable remote excitation of the acoustic wave from the bottom with a wide range of potential spatio-temporal profiles. To enable optimal excitation and measurement of wave propagation in these films, thin aluminum films ( ~ 30 nm thick) were vapor-deposited on both sides of the architected membrane. A detailed description of the microfabrication procedure is presented in the Supplementary Information Section 3.

### Experimental characterization

A photoacoustic pump-probe experiment was developed to resolve propagating elastic waves in our architected films. Excitation of the acoustic waves was achieved by an infrared pulsed laser beam (Coherent FLARE-NX, wavelength 1030 nm, pulse energy 500 μJ, pulse width 1 ns). The mechanism of photoacoustic excitation involves rapid thermal expansion of thin aluminum films deposited on the sample (acting as transducers), resulting in the propagation of an acoustic pulse through the film. The duration of this pulse is determined, in part, by the thickness of the film and its bulk elastic wave speed. Particle displacements were measured by a custom-built heterodyne interferometer. The heterodyne system uses a continuous wave laser (20 mW 633 nm) source with the reference branch frequency shifted by 80 MHz, using an Acousto-Optic Modulator (AOM: EQ Photonics 3080-120) driven by an RF driver (Gooch and Housego 3910). The sample branch of the interferometer was focused on the measurement point on the sample, using a 20× objective lens before being recombined and detected at a balanced photo detector (ThorLabs PDB 230A). Under ambient conditions, this generates an 80 MHz beat signal, which undergoes phase shifts due to the displacement of the measurement point. During the propagation of the acoustic wave, the time-resolved phase shift was measured using a lock-in amplifier (Zurich Instruments GHFLI) and read out into a high-speed digital oscilloscope (Tektronix MSO64B). Particle displacements are

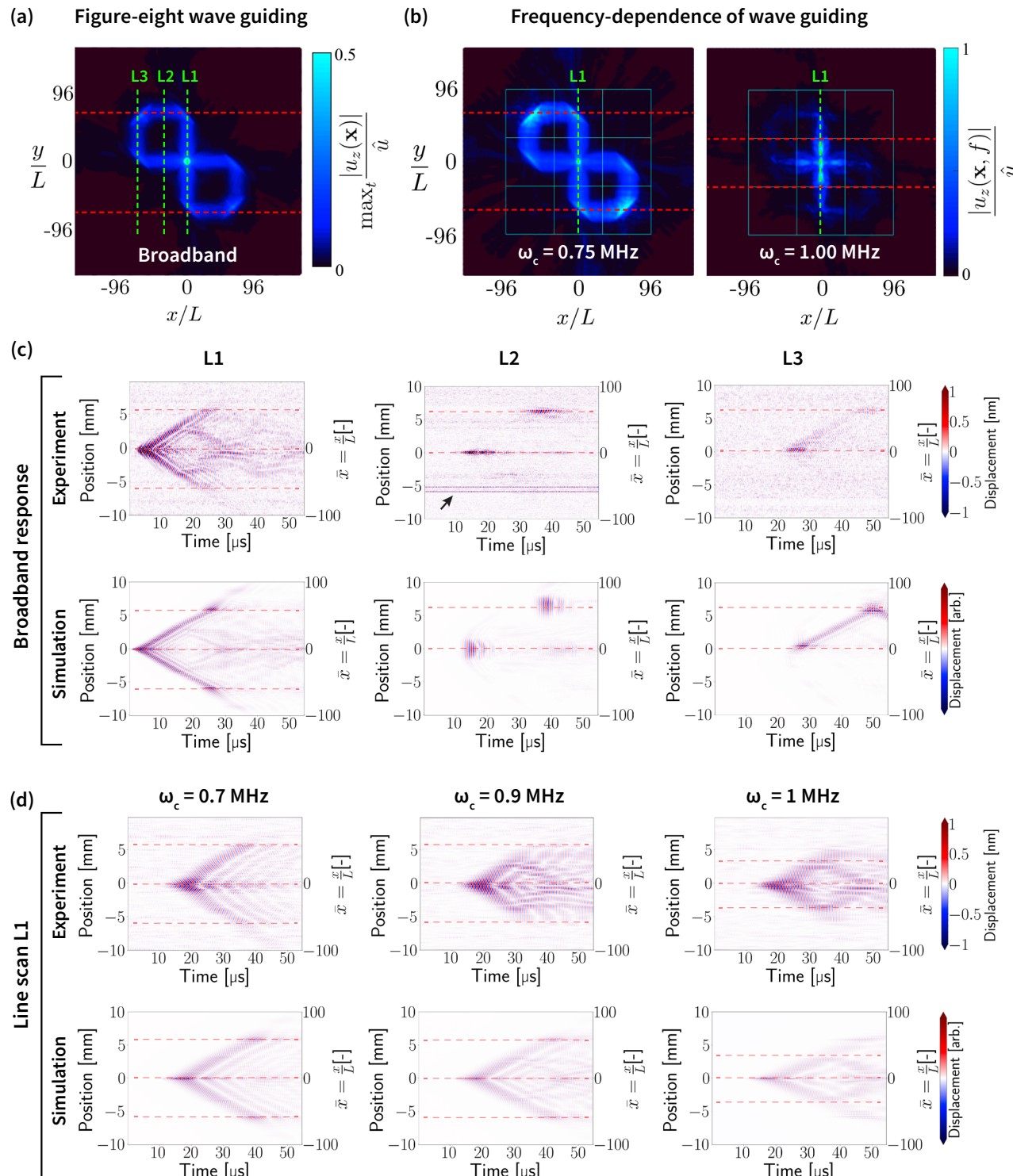

**Fig. 4 | Experimental results showing figure-eight wave guidance.** Finite element simulation results for (**a**) transient wave propagation and (**b**) frequency-dependent wave guiding in the graded architecture. Green dashed lines indicate the positions of line scans collected during experiments. **c** Experimental results from scans L1, L2, and L3, in comparison with equivalent data from finite element simulations, show

an excellent match. Red dashed lines indicate positions beyond which no wave propagation was measured/calculated (equivalent positions are shown in (**a**)). **d** Comparisons of frequency-dependent responses between experiments and simulations for L1 confirm agreement across a wide frequency range.

directly proportional to the measured phase shift ($\varphi_m$), and are calculated as

$$\delta(t) = \frac{\lambda}{2\pi}\varphi_m(t), \qquad (1)$$

where $\lambda$ is the wavelength of the laser source and $t$ is time. Scanning measurements were performed using two automated stages remotely controlled using Python code. At each measurement point, 50 time series data points were averaged to increase the signal-to-noise ratio. Further details regarding data analysis are summarized in Supplementary Information Sections 4 and 5.

## Data availability

The experimental data generated in this study have been deposited in the ETH Zurich Research Collection database, which is publicly available at https://doi.org/10.3929/ethz-c-000785222.

## Code availability

The codes for optimal tile design and assembly, and experimental data analysis generated in this study have been deposited in the ETH Zurich Research Collection database, which is publicly available at https://doi.org/10.3929/ethz-c-000785222.

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

## Acknowledgements
The authors thank Dr. Emil Bronstein for his assistance in setting up experimental scans. C.D. acknowledges partial support from an ETH Zurich Postdoctoral Fellowship. Mask writing and fabrication were performed in the cleanroom facility of the Binnig and Rohrer Nano-technology Center of IBM Zurich.

## Author contributions
C.D., V.K. and D.M.K. designed the research. C.D. performed computational modeling and design. V.K. designed the microfabrication method and experiments. V.K. and C.D. performed the experiments and analysis. U.D. supported the microfabrication efforts. C.D., V.K. and D.M.K. wrote the manuscript.

## Funding

## Competing interests
The authors declare no competing interests.
