## [Peer Review File · Nature Communications]

Graded phononic metamaterials based on scalable microfabrication and design

Corresponding Author: Professor Dennis Kochmann

Version 0:

Reviewer comments:

Reviewer #1

(Remarks to the Author)

This paper deals with the introduction of a framework for the scalable inverse design of spatially graded metamaterials for elastic wave guiding and a scalable microfabrication method. This framework enables the design and realization of complex waveguides with hundreds of thousands of unit cells, with the potential to expand these to millions without changing the protocol. The scalable design is achieved through optimization using a ray tracing model for waves in spatially graded beam lattices.

The goal of the paper to expand metamaterials' design with the aim of manufacturing large architectures spanning thousands to millions of unit cells is commendable and extremely timely.

Indeed, as the authors point out, assembling large metamaterials' structures necessarily minimizes boundary effects that occur when combining different metamaterials together or combining a metamaterial to more « classical » engineering materials. The problem of addressing pertinent boundary conditions between different metamaterials is here addressed by designing different unit cells that are « compatible », i.e. can be connected together without creating material discontinuity.

However, the sentence used by the authors « which truly dissolve the differences between materials and structures » seems a bit excessive, because in true applications a metamaterial will always be connected to other materials, so that macroscopic boundary effects cannot be totally « dissolved ». I would rather suggest to replace the word « dissolved » with a more precise sentence in which it is better explained that, increasing the size of the metamaterial's structure, the effect of the unavoidable boundary effects is indeed minimized and can be neglected in the targeted applications after a certain specimen's size.

As said, the goal of looking for scalable solutions is commendable in view of reducing computational time for large-scale structures. However, also other strategies are available to approach the challenge of designing large scale metamaterials' structures within reasonable computational times. Among them, the most explored ones are those connected to the use of macroscopic (homogenized) models, especially those of micromorphic type (see e.g., <https://doi.org/10.1016/j.jmps.2022.104995>). Recently, such kind of simplified and computationally affordable methods have also been complemented with the concept of « effective interface forces » in order to deal with the aforementioned challenge related to boundary effects that are unavoidable in finite-sized metamaterials structures (see e.g., <https://doi.org/10.1016/j.jmps.2024.105558>).

Indeed coupling the proposed scalability methods with effective (macroscopic) models could further widen the possibility of approaching realistic designs at large scales.

It would thus be beneficial if the potentialities of the proposed models could be enlightened in this more extended viewpoint (e.g. in the introduction).

The presentation of an experimental scalable method to truly manufacture large-scale metamaterial structures is also extremely important and meritorious.

In the light of the previous comments, I highly recommend the publication of the present paper in Nature Communication by

implementing only Minor Revisions.

(Remarks on code availability)

Reviewer #2

(Remarks to the Author)

In this manuscript, the authors claim to “present a framework for scalable inverse design of spatially graded metamaterials for elastic wave guiding, together with a scalable microfabrication method”. While the authors’ metamaterial design and experiment are well conducted and documented, in the opinion of the reviewer there is little novelty, and there would thus be minimal impact for the community. Similarly, the introduction is incomplete to the degree that, when combined with the claims of the authors, is misleading.

In the opinion of the referee, the claimed novelty could be broken down into two sub-claims, as follows.

1. Introducing a scalable microfabrication method for large numbers of metamaterial unit cells in high Q factor suspended membranes. Their method is not remotely new. To pick just one uncited example from ~9 years ago, is Ref. [Sledzinska, Marianna, et al. "Fabrication of phononic crystals on free-standing silicon membranes." *Microelectronic engineering* 149 (2016): 41-45]. In this reference, the authors use a similar technique to manufacture 12 different patterns of 40,000 to 100,000 unit cells in each pattern on a freestanding silicon membrane for the creation of phononic crystals. Indeed, it is the referee’s understanding that, particularly at the scales the authors conducted their experiments, this sort of microfabrication process is quite standard. Particular communities where this type of fabrication is used regularly is cavity optomechanics, MEMS, and NEMS, in the form of suspended silicon nitride membranes. In addition, it is unclear how this article’s process is any different from their (cited) preprint Ref. [32].

2. A “scalable” phononic crystal design method allowing for gradients. It is unclear how this contribution is meaningfully different than their previous (cited) publication Ref. [28] from 2022.

I do not see that putting these two contributions together, and testing the result with laser ultrasonics, is novel, particularly given the existence of their prior two manuscripts Ref. [28] and [32]. The acoustic response of suspended silicon membranes is so standard at this point, one would not expect any surprises compared to simulation.

For publication anywhere, an appropriate literature review should be conducted, and the claims revised for accuracy. As will be listed below, it seems every few sentences in the introduction one encounters an erroneous, misleading, or vague statement. Even after this is corrected, it seems there will be minimal novelty, such that a journal like *Scientific Reports* could be appropriate.

Erroneous or misleading statements in the introduction:

- “It is challenging to model, design, and manufacture large architectures spanning thousands to millions of unit cells.” “Challenging” is subjective, and meaningless as concerns novelty.

- “Solving the problem of scalability promises a significantly enlarged and widely untapped design space for programmable control of (meta-)material functionality to enable emerging engineering applications.” What does “widely untapped” precisely mean? What examples specifically will be enabled? What specific “emerging engineering applications” are envisioned? The key example of this paper is a “figure 8” waveguide. Many other methods have previously shown more complex (and one-way) waveguiding, such as in the field of topological insulators, e.g., Ref. [Ma, Jingwen, Xiang Xi, and Xiankai Sun. "Experimental demonstration of dual-band nanoelectromechanical valley-hall topological metamaterials." *Advanced Materials* 33.10 (2021): 2006521], using less unit cells.

- “However, modeling and design of non-periodic architectures are difficult to scale to large architectures with many unit cells since, unlike for periodic architectures, each unit cell must be modeled.” “Difficult” is subjective, and meaningless as concerns novelty. One could consider a comparison of runtime for related design methods.

- “Existing spatial grading design methods have relied on heavily restricted design spaces (e.g., linear gradings [10] and analytical solutions [11]) or restricting assumptions of long wavelengths to enable homogenization [12, 13].” The author’s previous publication Ref. [28] is an “Existing spatial grading design method”.

- “On the experimental side, the fabrication of metamaterials consisting of large numbers of unit cells is challenging. Scaling fabrication beyond tens of unit cells per dimension is extremely challenging, making it difficult to truly fabricate metamaterials rather than structures.” “Challenging” is subjective, and meaningless as concerns novelty.

- “At the current limits of 3D printing, architectures with hundreds of unit cells per dimension [14] are within reach, but such methods are limited to polymeric materials, which are not ideal for wave guiding due to material damping.” This is not true. Here is one example for non-polymer two-photon 3D printing [Kotz, Frederik, et al. "Two-photon polymerization of nanocomposites for the fabrication of transparent fused silica glass microstructures." *Advanced Materials* 33.9 (2021): 2006341.].

- Scalability is especially challenging for nano- and microscale manufacturing, where two-photon lithography [15–20] and its variants [21] can hardly scale beyond samples with tens of unit cells per dimension in a tractable build time." This is not true. For instance, Ref. [Zhang, Leran, et al. "High-throughput two-photon 3D printing enabled by holographic multi-foci high-speed scanning." *Nano Letters* 24.8 (2024): 2671-2679.] shows printing 384 unit cells using two-photon lithography in 4.2 seconds. Scaling this to 100,000 unit cells would result in a print time of 18 minutes.

- "Microfabrication of optical metamaterials has achieved better scalability, with two-dimensional architectures reaching over a billion unit cells, using electron beam lithography and atomic layer deposition [22, 23]; however, this is achieved by patterning features onto a substrate, rather than creating a free-standing material architecture." This statement is misleading, as the authors here do not show a billion unit cells. As stated above, there are prior examples with similar demonstrated numbers of unit cells as this work.

(Remarks on code availability)

Reviewer #3

(Remarks to the Author)

This manuscript introduces a scalable design and modeling methodology for gradient-based phononic metamaterials with very large lattices. To overcome computational challenges, dispersion relations of individual unit cell designs are combined with ray tracing of large lattice tiles. Individual tiles are then combined to achieve targeted wave guiding. This approach is demonstrated for 2D lattices consisting of beam structures for guiding of elastic waves and experimentally realized using SOI microfabrication.

****Referee assessment****

The paper is well written and comprehensively covers the design methodology, modeling approach, fabrication, and experimental characterization. Extensive supplementary material covers most details missing from the main manuscript. The proposed methodology is interesting, however limitations and performance of the ray-tracing are not discussed. A critical point is an inaccurate claim of novelty regarding SOI microfabrication. A revision is recommended.

This and some other shortcomings are elaborated below:

1. Introduction and terminology

The authors call their structures "phononic metamaterial" and use both terms "elastic waves" and "mechanical waves". While usage differs from field to field and group to group, commonly the more general term would be acoustic metamaterial and acoustic wave which implies frequency-dependence and which applies to any wave medium. Phononic MM and phononic crystals have been used for devices covering both air/fluid-borne sound and structure-borne sound. Elastic waves and often also mechanical waves refer to structure-borne sound, i.e. waves in solids. The term mechanical metamaterials is often also limited to structures with unusual static/non-wave mechanical behavior, e.g. displaying negative Poisson ratio when stretched/compressed. The authors have mixed references of MM for waves in fluids and solid, but their specific design applies only to elastic waves.

It is recommended to use acoustic waves and acoustic MM when referring to the general idea of dynamic metamaterials for sound waves (incl. ultrasound, infrasound, ...) to distinguish from mechanical metamaterials for non-waves. And to specify elastic MM and elastic waves when limited to solids.

2. Incorrect fabrication claims

In the introduction the authors write: "At the current limits of 3D printing, architectures with hundreds of unit cells per dimension [14] are within reach, but such methods are limited to polymeric materials". But 3D printing is not limited to polymers. Metal and ceramic printing has been available for over a decade and utilized in various acoustic metamaterials and phononic crystals, e.g.:

(2017) <https://doi.org/10.3390/ma10101125>

(2017) <https://doi.org/10.1016/j.ultras.2017.07.017>

(2019) <https://doi.org/10.1016/j.jsv.2019.115167>

More critically, later the authors write about the SOI microfabrication:

"The novelty of this microfabrication method is that free-standing architected films (as opposed to a pattern deposited on a substrate) are manufacturable in a scalable fashion."

Using SOI wafers to fabricate free-standing structured slabs of silicon, back-etching silicon substrates, or using sacrificial layer technology for free-standing elements is nothing new to the world of acoustic metamaterials and phononic crystals.

Early examples include, but are not limited to:

(2008) <https://doi.org/10.1016/j.sna.2007.10.081>

(2008) <https://doi.org/10.1063/1.2939097>

(2010) <https://doi.org/10.1021/nl102918q>

(2015) <https://doi.org/10.1016/j.mee.2015.09.004>

The authors need to temper their claims and acknowledge a long history of CMOS, MEMS, SOI-MEMS, etc. microfabrication in the fields of metamaterials.

Looking at photonic/optical metamaterials further broadens the existing literature. Incidentally, the introduction statement regarding optical MM of "rather than creating a free-standing material architecture" is also not accurate. There are countless examples of photonic crystals and MMs built on suspended membranes, beams, etc.

3. Ray-tracing vs 2D FE simulation

One of the main claims in this manuscript is the much more efficient modelling, where finite element models of the mechanical vibration is only used for eigenmode analysis of individual unit cells and calculation of the dispersion relation/band structure. The long range model uses ray tracing. However, the authors do not fully describe the limits of the ray tracing model. By its nature, ray tracing is a high-frequency/short-wavelength approximation instead of the full wave equation(s), here further limited to out-of-plane bending waves. It neglects any wave coupling or interaction with the structure. The authors need to justify this approach by at least mentioning the relevant wave speeds/wavelengths for their materials and designs. The dispersion relation in Fig. 1 is insufficiently detailed to infer this information.

While the experimental results appear to validate the ray-tracing, a comparison with a FEM wave equation result, maybe for a single tile, would be more convincing.

More importantly, the authors need to put the higher efficiency of their modeling approach into quantitative values. Time, memory, CPU/GPU cycles or similar measures should be given at least in the supplementary material. A computational resources comparison of full wave equation model vs ray-tracing for maybe a single tile would elevate the claims. This may also include computational quantification of the inverse design procedure.

4. Other medium/minor issues

- All tile examples show only 90° turns. Are other angles not possible? If not, why?
- Fig. 2: Font size becoming too small and hard to read. In general, font in figures should not be much smaller than in caption.
- Fabrication: "thin aluminum transducers (~ 30 nm thick)" are mentioned. Aluminum is not an active material and calling it a transducer like a piezoelectric material is strange. It appears to play the role of mirror/reflective surface for the acousto-optical transduction for excitation and detection.
- Fig. 3: The photo in d) has a scale bar but no mention of the scale in the caption.
- Fig. 4: Font size too small and hard to read.
- Fig. 4: order of a),c),b),d) is confusing.
- Methods section: Several aspects are repeated from previous text, esp. doubling information regarding the microfabrication p. 6-7 and 9-10.

(Remarks on code availability)

I have checked availability and type of code/files. Most calculation and data processing appears to use MATLAB but I have not reviewed the MATLAB code.

Dispersion relations calculated from the finite element wave equation/eigenmode model are only provided as results, not the models/FEM codes.

Many figure files are provided. However, experimental data is not sorted well and does not appear to include any file description or file naming scheme.

Version 1:

Reviewer comments:

Reviewer #1

(Remarks to the Author)

The comments have been addressed satisfactorily.

I thus recommend publication of the paper in its present form.

(Remarks on code availability)

Reviewer #2

(Remarks to the Author)

While the authors have made substantial replies in the rebuttal letter, I do not feel substantive improvements were made to the manuscript regarding modifying their arguments to accurately place their contribution in the context of existing and prior capabilities. Key references that would conflict with their arguments of novelty are still missing.

I do not believe this is appropriate to be published anywhere in its current form. Even if their description of the context of their contribution is improved, this work does not have sufficient novelty for Nature Communications. Scientific Reports or NJP Metamaterials could be alternate suitable venues.

Here is just one other, again uncited, example of using laser ultrasonics to measure phononic responses of patterned plates

[Zhao, Jinfeng, et al. "On-chip valley phononic crystal plates with graded topological interface." *International Journal of Mechanical Sciences* 227 (2022): 107460.]. In that reference they study ~1.7-1.9 MHz waves in a patterned silicon plate with patterned area stretching over 31x37 cm within a 4 inch diameter wafer.

The authors' focus in the rebuttal letter on subtle difference from articles brought to their attention by the Reviewers misses the point. Reviewers should not have to do the authors' literature review for them.

Below, I will also respond to the subset of their replies that I feel warrant particular response.

Author: "the main focus of this manuscript is the application of our experimental-computational framework to design of new elastic wave guides across micrometer to millimeter length scales, like the figure-8 case discussed."

I do not understand what the experimental part is in "experimental-computational framework". It seems the design is purely computational, then there is experimental validation. Neither the fabrication method nor the characterization method is substantially new.

In addition, I am unclear from the manuscript as to why someone would want to make these large gradient based structures, when the same thing can be done in a much smaller footprint. For instance, conventional (not even topological or phononic-crystal-based) elastic waveguides have been around for many decades. A broadband, figure 8 waveguide could be easily done in this conventional context via design of impedance mismatch and total internal reflection.

Author: In general, we stress that it was not our intention to claim that we have invented a new technique in microfabrication. We apologize if we inadvertently created that impression. Studies that have breached the nanoscale (like the one that the reviewer correctly pointed out) have achieved free-standing structures of hundreds of micrometers. In certain applications of mechanical wave guiding, however, it may be critical to guide waves over much longer distances; this is non-trivial to achieve.

What applications, specifically, are these critical for? This has still been left as nebulous in the manuscript.

Author: We have used existing microfabrication techniques to create elastic wave-guiding metamaterials across length scales spanning micrometers to millimeters – a niche range that we refer to as a "meso" scale. Our manuscript presents in detail the advantages and challenges of accessing this "intermediate" range.

What advantages and where are they stated? Do you mean "This admits controlling wave motion in the tens of kHz to MHz regime across centimeters – a regime of long-standing interest to the field". If so: i) this is still nebulous; ii) Someone controlling kHz to MHz waves would just use larger structures there is no need to use few micron beams (see paper cited above).

Author: For example, if we could fabricate nanoscale structures on a free-standing membrane spanning tens of millimeters (like ours), this would be a momentous step forward in scalable wave-guiding technology, e.g., for controlled design of the "long wavelength" dispersion response of mechanical waves in signal processing.

How is this "momentous" exactly? What would this enable in terms of new phenomenology or functionality? Why would one just not use larger unit cells (see paper cited above)?

Author: Our "meso" scales also allow measurements of the full spatio-temporally resolved wave propagation with sufficiently high resolution. Such high-dimensional experimental data are crucial to validate computational models, verify and realize complex waveguide designs obtained from optimization schemes, as demonstrated in this study.

I do not understand this comment. Laser ultrasonic characterization, including full field imaging has been around for decades (e.g. [Tachizaki, Takehiro, et al. "Scanning ultrafast Sagnac interferometry for imaging two-dimensional surface wave propagation." *Review of Scientific Instruments* 77.4 (2006)], or the paper cited above)

Author: Prior works in the MEMS and NEMS communities (which are clearly relevant here from a microfabrication perspective), to our knowledge have not explored large, free-standing microstructured thin films (spanning micrometers to millimeters)—owing, among others, to the prevalent interest in resonator technology and surface acoustic waves (SAWs), which do not require structures to be free-standing across a large spatial domain. However, such large, free-standing structures naturally enable the propagation of more complex wave modes (e.g., Lamb waves).

I do not understand this comment. Lamb waves are routinely considered in phononic crystal design – see, e.g., another, uncited, example [Hyun, Jaeyub, Wonjae Choi, and Miso Kim. "Gradient-index phononic crystals for highly dense flexural energy harvesting." *Applied Physics Letters* 115.17 (2019)].

Author: As a side note, Sledzinska et al. (2016) focused on stabilizing ultra-thin films, using a polymer layer. While a novel technique, it is unclear how waves can propagate in that configuration over long distances due to damping—especially when aiming for lower frequencies, as discussed here. Discrepancies due to damping may affect quantitative comparisons of the full spatio-temporal evolution of waves through those wave guides.

There are many suspended membrane studies that do not use polymer layers. Here is another, uncited, example [Graczykowski, Bartłomiej, et al. "Phonon dispersion in hypersonic two-dimensional phononic crystal membranes." *Physical Review B* 91.7 (2015): 075414].

Author: Finally, Sledzinska et al., (2016) and several others within this domain focused on improving the fabrication of nanoscale waveguides rather than spatial grading for pre-defined wave guiding capabilities. As has been evident from literature [41, 12, 14, 47], this is a non-trivial problem with recent progress at macroscopic scales (including our own prior work, e.g., [25, 15]). The present manuscript demonstrates significantly improved synergy between scalability in computation, fabrication, and measurement for more robust spatial grading strategies.

I do not understand this comment. Most other papers just treat this “synergy” as experimental validation of a computationally designed metamaterial.

Author: The presented large-scale waveguides are made possible by the “tile design” approach, which is introduced here for the first time (and nontrivial due to, among others, requirements of matched dispersion relations at tile interfaces, treatment of boundaries, and the design targets required in such tiles). Brute-force optimization of the entire domain based on [32] would not be computationally feasible for the designs considered in this manuscript.

There is no proof by the authors that this brute force optimization is not feasible, particularly given the used (and prior demonstrated in Ref. [24]) Ray tracing algorithm. Why is the tiling needed? There is no evidence.

Author: Ref. [24] exclusively optimized rays based on target locations (e.g., wave focusing). Here, we present for the first time how to optimize designs such that rays enter and exit perpendicular to the boundary (which is essential for the tiling to function). Ref. [24] only implements mass-spring network toy problems with analytical dispersion relations; the current manuscript's demonstration of beam networks with numerically computed dispersion relations is a major improvement that is key for enabling experimental demonstration.

These both seem like relatively incremental advances in the opinion of the reviewer.

Author: Thank you for pointing out that this wording is not precise. We have updated the wording to “Modeling, design, and manufacturing efficiency remains a bottleneck for architectures spanning tens of thousands to millions of unit cells...” We point out that the Springer-Nature journal family does not appreciate precise novelty claims (“we ask authors to refrain from making priority or novelty claims and to remove qualitative evaluations of their own work”; “avoid phrases like ‘for the first time’ and ‘unprecedented’”).

The statements the authors make must still be accurate. In the Reviewers opinion, given the prior references and related work in other fields, this is not a bottleneck as the authors have claimed.

Author: When referring to a “widely untapped design space”, we are specifically referring to spatially graded architectures with complicated spatial profiles spanning multiple length scales. As mentioned in paragraph 3 of the introduction, existing work on spatially graded phononic metamaterials has been restricted to simple grading profiles—such as grading along one dimension only, or radially symmetric patterns. Two-dimensional spatial gradings with complicated grading profiles spanning multiple length scales have not been previously considered. Scaling to large architectures with many unit cells creates the opportunity to explore gradings with more elaborate spatial structures. We are not tied to a particular application, as we present a fundamental framework for exploring what new capabilities are possible in a previously unexplored design space of complex, multiscale, 2D spatial gradings. However, we anticipate that expanding the design space and functionality of graded metamaterials will enhance applications that graded metamaterials (with simple grading

profiles) have shown promise for already, such as energy harvesting [10] and mechanical/acoustic signal processing [11], in addition to potentially enabling new applications that could benefit from enhanced control over wave guiding, localization, and focusing, such as microfluidics [65].

The authors still have not clearly stated how this would be beneficial over simpler solutions. They have not demonstrated how energy harvesting, acoustic signal processing, focusing, or microfluidics would be improved via this design, compared to simpler, existing approaches.

Author: We acknowledge that many approaches to wave guiding have been explored. Topological metamaterials, such as the example of (Ma et al., 2021), are appealing in many cases, e.g., due to their robustness to defects. One potential drawback is that they are restricted to often narrow bandgaps in the bulk, and to discrete wave guiding angles aligned with lattice interfaces. Similarly, many other approaches to acoustic and phononic metamaterials, which leverage bandgap design, are limited to narrow frequency bands, unlike the results presented here. The introduction has been updated to reflect these points.

Topological waveguides were just one example. There are many more examples of topologically trivial waveguides simply by virtue of effective elastic property mismatches, which do not have a bandwidth limitation.

Author: "Thank you for pointing out the lack of precision in this statement. We have modified these sentences to the following: "... Going from tens to hundreds of thousands of unit cells at the meso-scale requires the adaptation of specialized microfabrication techniques that have rarely been exploited for free-standing elastic wave guides [27] – especially spanning tens of micrometers to millimeters."

"Rarely" does not contribute to the scientific novelty of the work.

Author: Thank you for pointing out this incorrect statement. We were referring to conventional macroscale 3D printing techniques as enabled by commercially available machines, not microfabrication processes. In the context of microfabrication, we have added further references on the fabrication of non-polymeric materials to the revised manuscript, though those have the drawback that the trade-off between resolution and field of view still remains a challenge. In response to similar comments from Reviewer 3, we have also included the broader class of conventional metal and ceramic additive manufacturing in our statement. The sentence in question has been modified as follows: "The current state of the art in commercial macroscale 3D printing technology enables architectures with hundreds of unit cells per dimension [28]. These methods, which fall within the broad category of additive manufacturing (AM), have been widely exploited for polymeric materials, which unfortunately are not suitable for wave propagation because of material-intrinsic damping. Although exciting new directions in metal and ceramic AM have emerged [29, 30, 31], careful control of printing resolution and material properties still poses challenges. Microfabrication techniques enabled by multi-photon lithography, traditionally using polymeric materials at significantly higher spatial resolution [32], have overcome this limitation [33, 34]. Such techniques, while growing rapidly, are still nascent in their ability to achieve large sample sizes close to 100 mm while maintaining sub-micrometer spatial resolution [35, 36], lesser so when spatially-graded non-periodic architectures are involved."

I do not think the arguments added to the introduction here are accurate. Even commercial entities are now advertising "wafer scale" production through 3D printing, including the availability of silica resins [<https://www.nanoscribe.com/en/applications/wafer-scale-production-through-3d-direct-printing/>, <https://www.nanoscribe.com/en/products/gp-silica/>].

Author: While the overall scaling argument is true in principle, we kindly disagree with the reviewer on its applicability to the technique proposed in the the cited reference. It is crucial to take into account the technique itself. The scaling from 384 to 100,000 unit cells (for the same unit cell size) is non-trivial – the sample sizes are limited by the range of the galvanometer mirrors used in their setup (300 × 300 μm). To increase this range, the galvanometer scanning system must be coupled to a scanning stage. Hence, a simple linear time scaling from 384 to 100,000 unit cells is an unreasonable assumption. Such "stitching" methods have been shown to introduce defects in printed samples [33]. As mentioned before, while attempts at making such scalable samples are improving fast, the technology is still in the nascent/development stage. Finally, we reiterate that this technique has been shown to work for polymeric samples only as yet.

I assert that this is incorrect, see the reply to the prior statement.

(Remarks on code availability)

Reviewer #3

(Remarks to the Author)

The authors have revised their manuscript in a thorough manner and addressed most reviewer comments in an adequate fashion. The argumentation in regards to sufficient novelty is mostly convincing and the authors have tempered and modified incorrect/inaccurate claims.

However, in several responses the authors purely point to changes/additions in the supplementary materials. They should carefully consider whether statements/claims are needed in the main manuscript, considering that supplementary material is only viewed by a fraction of readers. One example would be the computational cost and gains of the ray tracing approach, which is not reflected in the main manuscript.

A note regarding the PDF manuscript, it is quite large, especially due to the figures. When viewing the manuscript in a browser, Fig. 4 does not load completely (it does load when viewing in Acrobat). In general, all figures load slowly and may even impede scrolling. The authors should make sure that figures (esp. pixel graphics) only use the resolution necessary for normal viewing and check their PDF in different viewers.

(Remarks on code availability)

Revision of NCOMMS-25-56068-T: “**Graded phononic metamaterials: scalable design meets scalable microfabrication**” by Charles Dorn, Vignesh Kannan, Ute Drechsler, and Dennis M. Kochmann

We thank the reviewers for their thoughtful and constructive comments. We have taken them as basis for the revision of our manuscript.

In the following, please allow us to respond to the reviewer’s questions and comments – explaining the changes made in our revised manuscript (which are highlighted in red in the attached manuscript). We refer to the reference numbering of the revised manuscript, which is copied below.

Kindly note that we discovered a small mistake in our data analysis code, resulting in a minor error of $\sim 1\%$ in the displacement magnitude. This has been corrected in the revised manuscript.

Reviewer 1:

This paper deals with the introduction of a framework for the scalable inverse design of spatially graded metamaterials for elastic wave guiding and a scalable microfabrication method. This framework enables the design and realization of complex waveguides with hundreds of thousands of unit cells, with the potential to expand these to millions without changing the protocol. The scalable design is achieved through optimization using a ray tracing model for waves in spatially graded beam lattices.

The goal of the paper to expand metamaterials’ design with the aim of manufacturing large architectures spanning thousands to millions of unit cells is commendable and extremely timely.

Indeed, as the authors point out, assembling large metamaterials’ structures necessarily minimizes boundary effects that occur when combining different metamaterials together or combining a metamaterial to more “classical” engineering materials. The problem of addressing pertinent boundary conditions between different metamaterials is here addressed by designing different unit cells that are “compatible”, i.e. can be connected together without creating material discontinuity.

However, the sentence used by the authors “which truly dissolve the differences between materials and structures” seems a bit excessive, because in true applications a metamaterial will always be connected to other materials, so that macroscopic boundary effects cannot be totally “dissolved”. I would rather suggest to replace the word “dissolved” with a more precise sentence in which it is better explained that, increasing the size of the metamaterial’s structure, the effect of the unavoidable boundary effects is indeed minimized and can be neglected in the targeted applications after a certain specimen’s size.

Response: The reviewer makes a valid point, which is why we have modified the quoted statement in the revised manuscript. We only “aim at dissolving the differences between materials and structures”, though this can never be “truly” achieved, unless when dealing with an infinite medium. In the reported system, we approach as close as possible the separation of scales between micro- and macroscale (though a true scale separation is out of reach).

Reviewer 1: *As said, the goal of looking for scalable solutions is commendable in view of reducing computational time for large-scale structures. However, also other strategies are available to approach the challenge of designing large scale metamaterials’ structures within reasonable computational times. Among them, the most explored ones are those connected to the use of macroscopic (homogenized) models, especially those of micromorphic type (see e.g., <https://doi.org/10.1016/j.jmps.2022.104995>). Recently, such kind of simplified and computationally affordable methods have also been complemented with the concept of “effective interface forces” in order to deal with the aforementioned challenge related to boundary effects that are unavoidable in finite-sized metamaterials structures (see e.g., <https://doi.org/10.1016/j.jmps.2024.105558>).*

Indeed coupling the proposed scalability methods with effective (macroscopic) models could further widen the possibility of approaching realistic designs at large scales.

It would thus be beneficial if the potentialities of the proposed models could be enlightened in this more extended viewpoint (e.g. in the introduction).

Response: Thank you for pointing out micromorphic models as a potentially useful concept in pursuit of efficient and scalable metamaterial modeling. We agree that micromorphic models, among other low- and finite-frequency homogenization methods, could be very useful in certain settings for certain design objectives. We point out a key distinction between ray tracing and typical homogenized models: ray tracing uses the entire dispersion relations (under the assumption that they are “locally” representative), while finite-frequency homogenization methods provide means for dispersion relation approximation. Thus, high-frequency homogenization methods may be particularly useful for cases when the dispersion relation itself is expensive to compute, providing an efficient alternative for, e.g., bandgap design. Alternatively, ray tracing is particularly suitable for the case studied in this paper – given a unit cell design space and the corresponding dispersion relations, design a spatial grading for wave guidance. We are not aware of micromorphic models that accurately take into account general spatial variations of the unit cell geometry (this would likely require homogenization on the fly, which is prohibitively expensive), which is required for the design optimization presented here. Moreover, the type of optimization targets presented here are nontrivial to realize in a classical continuum setting (e.g., making waves turn by 90° without imposing the exact timing of the wave), which is why we use ray tracing. We have updated our discussion of homogenization-based methods in the introduction accordingly.

Reviewer 1: *The presentation of an experimental scalable method to truly manufacture large-scale metamaterial structures is also extremely important and meritorious.*

In the light of the previous comments, I highly recommend the publication of the present paper in Nature Communication by implementing only Minor Revisions.

Response: We thank the reviewer for the positive feedback.

Reviewer 2:

In this manuscript, the authors claim to “present a framework for scalable inverse design of spatially graded metamaterials for elastic wave guiding, together with a scalable microfabrication method”. While the authors’ metamaterial design and experiment are well conducted and documented, in the opinion of the reviewer there is little novelty, and there would thus be minimal impact for the community. Similarly, the introduction is incomplete to the degree that, when combined with the claims of the authors, is misleading.

In the opinion of the referee, the claimed novelty could be broken down into two sub-claims, as follows.

- 1. Introducing a scalable microfabrication method for large numbers of metamaterial unit cells in high Q factor suspended membranes. Their method is not remotely new. To pick just one uncited example from 9 years ago, is Ref. [Sledzinska, Marianna, et al. “Fabrication of phononic crystals on free-standing silicon membranes.” *Microelectronic engineering* 149 (2016): 41-45]. In this reference, the authors use a similar technique to manufacture 12 different patterns of 40,000 to 100,000 unit cells in each pattern on a freestanding silicon membrane for the creation of phononic crystals. Indeed, it is the referee’s understanding that, particularly at the scales the authors conducted their experiments, this sort of microfabrication process is quite standard. Particular communities where this type of fabrication is used regularly is cavity optomechanics, MEMS, and NEMS, in the form of suspended silicon nitride membranes. In addition, it is unclear how this article’s process is any different from their (cited) preprint Ref. [32].*

Response: We thank the reviewer for this critical and very helpful feedback. First, we would like to state

that this manuscript does not purely focus on the microfabrication technique, but also on inverse computational design and detailed full-field spatio-temporal characterization of the wave through our fabricated structure. In this sense, the main focus of this manuscript is the application of our experimental-computational framework to design of new elastic wave guides across micrometer to millimeter length scales, like the figure-8 case discussed. This is also the main difference between the cited preprint, ref. [32], which focuses on the development of the fabrication and experimental techniques and protocol that form this framework. We note that the two manuscripts are designed to complement each other ([32] does not constitute prior work but was submitted in parallel to this manuscript). With this clarification, allow us to address the specific criticism regarding fabrication.

In general, we stress that it was not our intention to claim that we have invented a new technique in microfabrication. We apologize if we inadvertently created that impression. Studies that have breached the nanoscale (like the one that the reviewer correctly pointed out) have achieved free-standing structures of hundreds of micrometers. In certain applications of mechanical wave guiding, however, it may be critical to guide waves over much longer distances; this is non-trivial to achieve. We have used existing microfabrication techniques to create elastic wave-guiding metamaterials across length scales spanning micrometers to millimeters – a niche range that we refer to as a “meso” scale. Our manuscript presents in detail the advantages and challenges of accessing this “intermediate” range.

The reviewer’s statement that the number of unit cells reported in Sledzinska et al. (2016) are similar to ours is indeed true. However, our work distinguishes itself from the afore-cited reference (and others in the MEMS and NEMS literature) through the following aspects:

1. The cited reference developed techniques to create nanoscale structures (with maximum ranges of $100 \times 100 \mu\text{m}$) with target frequencies in the GHz regime. Our objective was to create membranes that target the frequency range of 100’s of kHz to tens of MHz. Fabrication of such high densities of unit cells at our “meso” scales in free-standing membranes spanning tens of millimeters are not as common but have powerful applications in mechanical wave-guiding at those frequency ranges. Multiple studies in the mechanics literature [2, 64, 12, 41] seek to target such scales (and above), but have not tapped into the microfabrication routes exploited here. Hence, the realization of such large waveguides has not been reported yet to the best of our knowledge.
2. The development of large free-standing wave guides paves the way for critical technology requiring the propagation of waves across large physical distances (irrespective of the microstructural length scales). For example, if we could fabricate nanoscale structures on a free-standing membrane spanning tens of millimeters (like ours), this would be a momentous step forward in scalable wave-guiding technology, e.g., for controlled design of the “long wavelength” dispersion response of mechanical waves in signal processing. We believe that our microscale structures are a significant step in this direction.
3. Our “meso” scales also allow measurements of the full spatio-temporally resolved wave propagation with sufficiently high resolution. Such high-dimensional experimental data are crucial to validate computational models, verify and realize complex waveguide designs obtained from optimization schemes, as demonstrated in this study. This would be significantly more difficult to achieve in nanoscale phononic waveguides.
4. Prior works in the MEMS and NEMS communities (which are clearly relevant here from a microfabrication perspective), to our knowledge have not explored *large, free-standing* microstructured thin films (spanning micrometers to millimeters)—owing, among others, to the prevalent interest in resonator technology and surface acoustic waves (SAWs), which do not require structures to be free-standing across a large spatial domain. However, such large, free-standing structures naturally enable the propagation of more complex wave modes (e.g., Lamb waves).
5. As a side note, Sledzinska et al. (2016) focused on stabilizing ultra-thin films, using a polymer layer. While a novel technique, it is unclear how waves can propagate in that configuration over long distances due to damping—especially when aiming for lower frequencies, as discussed here. Discrepancies due to damping may affect quantitative comparisons of the full spatio-temporal evolution of waves through those wave guides.

6. Finally, Sledzinska et al., (2016) and several others within this domain focused on improving the fabrication of nanoscale waveguides rather than spatial grading for pre-defined wave guiding capabilities. As has been evident from literature [41, 12, 14, 47], this is a non-trivial problem with recent progress at macroscopic scales (including our own prior work, e.g., [25, 15]). The present manuscript demonstrates significantly improved synergy between scalability in computation, fabrication, and measurement for more robust spatial grading strategies.

To address the reviewer’s comments and to clarify the above in our manuscript, we have made modifications in the Section *Microfabrication: Silicon wafers to beam-based metamaterials*, now stating the following: “We exploit the state of the art in microfabrication technology to manufacture scalable phononic waveguides with micrometer feature resolution of sizes up to 80 mm in diameter. Microfabrication technology has pushed resolution limits down to the nanometer scale [32, 52, 53, 54, 55], including significant advances from the fields of micro- and nano-electromechanical devices (MEMS/NEMS) [56, 57, 58, 59, 60, 27]. Most of those studies focused on devices bonded to a substrate or free-standing “nano-architectures” no larger than hundreds of micrometers. Using the aforementioned well-established techniques, we target a “mesoscale”, where the functional unit cells span micrometers and the free-standing structure spans almost a full silicon wafer. At these scales, our computational-fabrication-experimental framework achieves robust control of spatially-graded architecture. This admits controlling wave motion in the tens of kHz to MHz regime across centimeters – a regime of long-standing interest to the field [13, 25, 16, 61, 26].”

Reviewer 2: 2. A “scalable” phononic crystal design method allowing for gradients. It is unclear how this contribution is meaningfully different than their previous (cited) publication Ref. [28] from 2022. I do not see that putting these two contributions together, and testing the result with laser ultrasonics, is novel, particularly given the existence of their prior two manuscripts Ref. [28] and [32]. The acoustic response of suspended silicon membranes is so standard at this point, one would not expect any surprises compared to simulation.

Response: We kindly disagree with the reviewer. First, Ref. [51] (Ref. [32] in the original submission) is not prior work but was submitted in parallel to this manuscript (here, focusing on the scalable design and wave-guidance, there on experimental techniques). [51] is being reviewed by other experts in the field (focusing on experimental methods). Therefore, [51] does not infringe on the novelty or originality of this submission.

Second, Ref. [24] (Ref. [28] in the original submission) does indeed constitute relevant prior work, as it forms the theoretical basis for the optimization framework presented here, but there are many scientific advances made between Ref. [24] and the present manuscript:

1. The presented large-scale waveguides are made possible by the “tile design” approach, which is introduced here for the first time (and nontrivial due to, among others, requirements of matched dispersion relations at tile interfaces, treatment of boundaries, and the design targets required in such tiles). Brute-force optimization of the entire domain based on [32] would not be computationally feasible for the designs considered in this manuscript.
2. Ref. [24] exclusively optimized rays based on target locations (e.g., wave focusing). Here, we present for the first time how to optimize designs such that rays enter and exit perpendicular to the boundary (which is essential for the tiling to function).
3. Ref. [24] only implements mass-spring network toy problems with analytical dispersion relations; the current manuscript’s demonstration of beam networks with numerically computed dispersion relations is a major improvement that is key for enabling experimental demonstration.

Reviewer 2: For publication anywhere, an appropriate literature review should be conducted, and the claims revised for accuracy. As will be listed below, it seems every few sentences in the introduction one encounters an erroneous, misleading, or vague statement. Even after this is corrected, it seems there will be minimal novelty, such that a journal like *Scientific Reports* could be appropriate.

Response: We apologize for any inaccurate claims in our literature review and admit that some of our claims may have inadvertently been misplaced. We have made the necessary changes in our literature review in the revised manuscript. However, we do believe, based on the above responses and the changes made in the revised manuscript, that the novelty and contribution of our manuscript to the field of phononic metamaterials still holds weight and warrants publication. We hope our responses to the reviewer sufficiently highlight all the novel aspects of this work.

Reviewer 2: *Erroneous or misleading statements in the introduction:*

“It is challenging to model, design, and manufacture large architectures spanning thousands to millions of unit cells.” “Challenging” is subjective, and meaningless as concerns novelty.

Response: Thank you for pointing out that this wording is not precise. We have updated the wording to “Modeling, design, and manufacturing efficiency remains a bottleneck for architectures spanning tens of thousands to millions of unit cells...” We point out that the Springer-Nature journal family does not appreciate precise novelty claims (“we ask authors to refrain from making priority or novelty claims and to remove qualitative evaluations of their own work”; “avoid phrases like ‘for the first time’ and ‘unprecedented’”).

Reviewer 2: *“Solving the problem of scalability promises a significantly enlarged and widely untapped design space for programmable control of (meta-)material functionality to enable emerging engineering applications.” What does “widely untapped” precisely mean? What examples specifically will be enabled? What specific “emerging engineering applications” are envisioned? The key example of this paper is a “figure 8” waveguide. Many other methods have previously shown more complex (and one-way) waveguiding, such as in the field of topological insulators, e.g., Ref. [Ma, Jingwen, Xiang Xi, and Xiankai Sun. “Experimental demonstration of dual-band nanoelectromechanical valley-hall topological metamaterials.” *Advanced Materials* 33.10 (2021): 2006521], using less unit cells.*

Response: When referring to a “widely untapped design space”, we are specifically referring to spatially graded architectures with complicated spatial profiles spanning multiple length scales. As mentioned in paragraph 3 of the introduction, existing work on spatially graded phononic metamaterials has been restricted to simple grading profiles—such as grading along one dimension only, or radially symmetric patterns. Two-dimensional spatial gradings with complicated grading profiles spanning multiple length scales have not been previously considered. Scaling to large architectures with many unit cells creates the opportunity to explore gradings with more elaborate spatial structures.

We are not tied to a particular application, as we present a fundamental framework for exploring what new capabilities are possible in a previously unexplored design space of complex, multiscale, 2D spatial gradings. However, we anticipate that expanding the design space and functionality of graded metamaterials will enhance applications that graded metamaterials (with simple grading profiles) have shown promise for already, such as energy harvesting [10] and mechanical/acoustic signal processing [11], in addition to potentially enabling new applications that could benefit from enhanced control over wave guiding, localization, and focusing, such as microfluidics [65].

We acknowledge that many approaches to wave guiding have been explored. Topological metamaterials, such as the example of (Ma et al., 2021), are appealing in many cases, e.g., due to their robustness to defects. One potential drawback is that they are restricted to often narrow bandgaps in the bulk, and to discrete wave guiding angles aligned with lattice interfaces. Similarly, many other approaches to acoustic and phononic metamaterials, which leverage bandgap design, are limited to narrow frequency bands, unlike the results presented here.

The introduction has been updated to reflect these points.

Reviewer 2: *“However, modeling and design of non-periodic architectures are difficult to scale to large architectures with many unit cells since, unlike for periodic architectures, each unit cell must be modeled.” “Difficult” is subjective, and meaningless as concerns novelty. One could consider a comparison of runtime for related design methods.*

Response: Thank you for pointing out the lack of precision in this statement. We have updated this section of the introduction to have a more specific discussion of the challenges of modeling large-scale architectures.

To model finite-frequency (i.e., above the homogenization limit) wave propagation through spatially graded metamaterials, the options are limited to high-fidelity simulations, where each unit cell is included in the model, or a ray tracing-based model. A computation time comparison between forward modeling via high-fidelity transient dynamic finite element simulations and ray tracing has been added in Supplementary Material Section 2.3, where the finite element simulations took on the order of hours, while the ray tracing simulations took on the order of seconds.

Reviewer 2: *“Existing spatial grading design methods have relied on heavily restricted design spaces (e.g., linear gradings [10] and analytical solutions [11]) or restricting assumptions of long wavelengths to enable homogenization [12, 13].” The author’s previous publication Ref. [28] is an “Existing spatial grading design method”.*

Response: The reviewer makes a fair point that the authors’ previous work, Ref. [24] (Ref. [28] in the original submission), is indeed an existing spatial grading design method. We emphasize that, while Ref. [24] is not restricted to simple gradings or long wavelengths, the current work is a distinct improvement with new capabilities not possible with the formulations of Ref. [24]. The key improvements are as follows:

- The current work, unlike Ref. [24], includes arbitrary exit curves and the capacity to design the wave vector at which the ray exits. These lead to a different form of the cost function and its gradient. These key differences are discussed in the Supplementary Material Section 1.2.
- The current work employs a modular design approach by designing a set of compatible tiles that can be assembled, which is crucial for pushing the limits of scalability. Design of compatible tiles is not possible unless the wave vector is prescribed at the tile boundary, hence modular design is not possible via Ref. [24]. Applying the design approach of Ref. [24] to the entire domain in the examples presented here is prohibitively expensive to scale to architectures of the size studied here. Therefore, the tile approach makes such design optimization possible.
- Ref. [24] only implements mass-spring network toy problems with analytical dispersion relations; the current manuscript’s demonstration of beam networks with numerically computed dispersion relations is a major improvement that is key for enabling experimental demonstration.

We have updated the introduction to have more precise wording and mention these key differences.

Reviewer 2: *“On the experimental side, the fabrication of metamaterials consisting of large numbers of unit cells is challenging. Scaling fabrication beyond tens of unit cells per dimension is extremely challenging, making it difficult to truly fabricate metamaterials rather than structures.” “Challenging” is subjective, and meaningless as concerns novelty.*

Response: Thank you for pointing out the lack of precision in this statement. We have modified these sentences to the following: “On the experimental side, the fabrication of 2D wave-guiding metamaterials, using standard machining and 3D-printing techniques, are limited to tens to hundreds of unit cells per dimension [25, 16, 26]. Going from tens to hundreds of thousands of unit cells at the meso-scale requires the adaptation of specialized microfabrication techniques that have rarely been exploited for free-standing elastic wave guides [27] – especially spanning tens of micrometers to millimeters.”

We point out again that we have made these changes also keeping in mind that the Springer-Nature journal family does not appreciate precise novelty claims in submitted manuscripts.

Reviewer 2: *“At the current limits of 3D printing, architectures with hundreds of unit cells per dimension*

[14] are within reach, but such methods are limited to polymeric materials, which are not ideal for wave guiding due to material damping.” This is not true. Here is one example for non-polymer two-photon 3D printing [Kotz, Frederik, et al. “Two-photon polymerization of nanocomposites for the fabrication of transparent fused silica glass microstructures.” *Advanced Materials* 33.9 (2021): 2006341.].

Response: Thank you for pointing out this incorrect statement. We were referring to conventional macroscale 3D printing techniques as enabled by commercially available machines, not microfabrication processes. In the context of microfabrication, we have added further references on the fabrication of non-polymeric materials to the revised manuscript, though those have the drawback that the trade-off between resolution and field of view still remains a challenge. In response to similar comments from Reviewer 3, we have also included the broader class of conventional metal and ceramic additive manufacturing in our statement.

The sentence in question has been modified as follows: “The current state of the art in commercial macroscale 3D printing technology enables architectures with hundreds of unit cells per dimension [28]. These methods, which fall within the broad category of additive manufacturing (AM), have been widely exploited for polymeric materials, which unfortunately are not suitable for wave propagation because of material-intrinsic damping. Although exciting new directions in metal and ceramic AM have emerged [29, 30, 31], careful control of printing resolution and material properties still poses challenges. Microfabrication techniques enabled by multi-photon lithography, traditionally using polymeric materials at significantly higher spatial resolution [32], have overcome this limitation [33, 34]. Such techniques, while growing rapidly, are still nascent in their ability to achieve large sample sizes close to 100 mm while maintaining sub-micrometer spatial resolution [35, 36], lesser so when spatially-graded non-periodic architectures are involved.”

Reviewer 2: *Scalability is especially challenging for nano- and microscale manufacturing, where two-photon lithography [15–20] and its variants [21] can hardly scale beyond samples with tens of unit cells per dimension in a tractable build time.” This is not true. For instance, Ref. [Zhang, Leran, et al. “High-throughput two-photon 3D printing enabled by holographic multi-foci high-speed scanning.” *Nano Letters* 24.8 (2024): 2671-2679.] shows printing 384 unit cells using two-photon lithography in 4.2 seconds. Scaling this to 100,000 unit cells would result in a print time of 18 minutes.*

Response: While the overall scaling argument is true in principle, we kindly disagree with the reviewer on its applicability to the technique proposed in the the cited reference. It is crucial to take into account the technique itself:

- The scaling from 384 to 100,000 unit cells (for the same unit cell size) is non-trivial – the sample sizes are limited by the range of the galvanometer mirrors used in their setup ($300 \times 300 \mu\text{m}$).
- To increase this range, the galvanometer scanning system must be coupled to a scanning stage. Hence, a simple linear time scaling from 384 to 100,000 unit cells is an unreasonable assumption.
- Such “stitching” methods have been shown to introduce defects in printed samples [33]. As mentioned before, while attempts at making such scalable samples are improving fast, the technology is still in the nascent/development stage.
- Finally, we reiterate that this technique has been shown to work for polymeric samples only as yet.

However, we realize that the statement cited by the reviewer may be misleading if read out of context. Hence, we have made a slight modification as follows: “Two-photon lithography [15–20] and its variants [21] demonstrate excellent accuracy and repeatability for samples with tens to hundreds of unit cells per dimension in a tractable build time. Recent modifications of the technique have shown significant improvement of fabrication time scales [43], albeit for polymeric base materials. Scaling those techniques to large sample sizes well beyond hundreds of micrometers has remained a challenge.”

Reviewer 2: *“Microfabrication of optical metamaterials has achieved better scalability, with two-dimensional architectures reaching over a billion unit cells, using electron beam lithography and atomic layer deposition [22, 23]; however, this is achieved by patterning features onto a substrate, rather than creating a free-standing material architecture.” This statement is misleading, as the authors here do not show a billion unit cells. As stated above, there are prior examples with similar demonstrated numbers of unit cells as this work.*

Response: We are afraid the reviewer may have misread the quoted statement from our manuscript. We write that microfabrication has already achieved over a billion unit cells in the scientific literature (by patterning features onto a substrate), for which we also provide references. We do not claim anywhere in our manuscript (and in particular not in those sentences) that we achieve over a billion unit cells with our samples.

Reviewer 3: *This manuscript introduces a scalable design and modeling methodology for gradient-based phononic metamaterials with very large lattices. To overcome computational challenges, dispersion relations of individual unit cell designs are combined with ray tracing of large lattice tiles. Individual tiles are then combined to achieve targeted wave guiding. This approach is demonstrated for 2D lattices consisting of beam structures for guiding of elastic waves and experimentally realized using SOI microfabrication.*

The paper is well written and comprehensively covers the design methodology, modeling approach, fabrication, and experimental characterization. Extensive supplementary material covers most details missing from the main manuscript. The proposed methodology is interesting, however limitations and performance of the ray-tracing are not discussed. A critical point is an inaccurate claim of novelty regarding SOI microfabrication. A revision is recommended. This and some other shortcomings are elaborated below:

1. *Introduction and terminology* The authors call their structures “phononic metamaterial” and use both terms “elastic waves” and “mechanical waves”. While usage differs from field to field and group to group, commonly the more general term would be acoustic metamaterial and acoustic wave which implies frequency-dependence and which applies to any wave medium. Phononic MM and phononic crystals have been used for devices covering both air/fluid-borne sound and structure-borne sound. Elastic waves and often also mechanical waves refer to structure-borne sound, i.e. waves in solids. The term mechanical metamaterials is often also limited to structures with unusual static/non-wave mechanical behavior, e.g. displaying negative Poisson ratio when stretched/compressed. The authors have mixed references of MM for waves in fluids and solid, but their specific design applies only to elastic waves. It is recommended to use acoustic waves and acoustic MM when referring to the general idea of dynamic metamaterials for sound waves (incl. ultrasound, infrasound, ...) to distinguish from mechanical metamaterials for non-waves. And to specify elastic MM and elastic waves when limited to solids.

Response: We agree that terminology varies throughout the literature and is a common point of ambiguity. We have updated the manuscript to consistently refer to “elastic waves” instead of “mechanical waves”. However, we believe it is justified to keep our terminology as “phononic metamaterials” rather than “acoustic metamaterials”. Our reasoning is that many (but not all) authors use “acoustic metamaterials” specifically to refer to architectures whose response is dominated by local resonance effects, e.g., in [2, 9, 63]. In our case, our ray tracing formulation is agnostic to whether local resonance and/or Bragg scattering dominates the response. In fact, in graded architectures it is conceivable to have local resonance behavior only in certain spatial regions. We think that the terminology “phononic metamaterials” is the most appropriate to the cases studied here, though we acknowledge that the terminology is not consistent throughout the literature.

Reviewer 3: 2. *Incorrect fabrication claims. In the introduction the authors write: “At the current limits of 3D printing, architectures with hundreds of unit cells per dimension [14] are within reach, but such methods are limited to polymeric materials”. But 3D printing is not limited to polymers. Metal and ceramic printing*

has been available for over a decade and utilized in various acoustic metamaterials and phononic crystals, e.g.:

(2017) <https://doi.org/10.3390/ma10101125>

(2017) <https://doi.org/10.1016/j.ultras.2017.07.017>

(2019) <https://doi.org/10.1016/j.jsv.2019.115167>

Response: We thank the reviewer for correcting us, and apologize for missing the broader class of AM studies. In response to similar comments from Reviewer 2, we have also included the smaller but growing class of microfabrication techniques for metallic and ceramic materials. The sentence has been modified in the revised manuscript as follows: “The current state of the art in commercial macroscale 3D printing technology enables architectures with hundreds of unit cells per dimension [28]. These methods, which fall within the broad category of additive manufacturing (AM), have been widely exploited for polymeric materials, which unfortunately are not suitable for wave propagation because of material-intrinsic damping. Although exciting new directions in metal and ceramic AM have emerged [29, 30, 31], careful control of printing resolution and material properties still poses challenges. Microfabrication techniques enabled by multi-photon lithography, traditionally using polymeric materials at significantly higher spatial resolution [32], have overcome this limitation [33, 34]. Such techniques, while growing rapidly, are still nascent in their ability to achieve large sample sizes close to 100 mm while maintaining sub-micrometer spatial resolution [35, 36].”

Reviewer 3: *More critically, later the authors write about the SOI microfabrication: “The novelty of this microfabrication method is that free-standing architected films (as opposed to a pattern deposited on a substrate) are manufacturable in a scalable fashion.” Using SOI wafers to fabricate free-standing structured slabs of silicon, back-etching silicon substrates, or using sacrificial layer technology for free-standing elements is nothing new to the world of acoustic metamaterials and phononic crystals. Early examples include, but are not limited to:*

(2008) <https://doi.org/10.1016/j.sna.2007.10.081>

(2008) <https://doi.org/10.1063/1.2939097>

(2010) <https://doi.org/10.1021/nl102918q>

(2015) <https://doi.org/10.1016/j.mee.2015.09.004>

Response: We thank the reviewer for this critical feedback. This critique is well received (and also in line with some of the points raised by Reviewer 2, which is why our response here is similar). We stress that it was not our intention to claim that we have invented a new microfabrication technique. We apologize if we inadvertently created that impression. Studies that have reached the nanoscale (like the ones that the reviewer correctly pointed out) have indeed achieved free-standing structures of hundreds of micrometers. In certain applications of mechanical wave guiding, however, it may be critical to guide waves over much longer distances; this is non-trivial to achieve. We have used existing microfabrication techniques to create elastic wave-guiding metamaterials across length scales spanning micrometers to millimeters – a niche range that we refer to as a “meso” scale. Moreover, the waveguides presented here require not periodic but spatially graded designs. Our manuscript presents in detail the advantages and challenges of accessing this “intermediate” range.

The revised manuscript now contains the following explanation: “We exploit the state of the art in microfabrication technology to manufacture scalable phononic waveguides with micrometer feature resolution of sizes up to 80 mm in diameter. Microfabrication technology has pushed resolution limits down to the nanometer scale [32, 52, 53, 54, 55], including significant advances from the fields of micro- and nano-electromechanical devices (MEMS/NEMS) [56, 57, 58, 59, 60, 27]. Most of those studies focused on devices bonded to a substrate or free-standing “nano-architectures” no larger than hundreds of micrometers. Using the aforementioned well-established techniques, we target a “mesoscale”, where the functional unit cells span micrometers and the free-standing structure spans almost a full silicon wafer. At these scales, our computational-fabrication-experimental framework achieves robust control of spatially-graded architecture. This admits controlling wave motion in the tens of kHz to MHz regime across centimeters – a regime of long-standing interest to the field [13, 25, 16, 61, 26].”

Reviewer 3: *The authors need to temper their claims and acknowledge a long history of CMOS, MEMS, SOI-MEMS, etc. microfabrication in the fields of metamaterials. Looking at photonic/optical metamaterials further broadens the existing literature. Incidentally, the introduction statement regarding optical MM of “rather than creating a free-standing material architecture” is also not accurate. There are countless examples of photonic crystals and MMs built on suspended membranes, beams, etc.*

Response: We hope that the aforementioned modifications in the revised manuscript have also addressed this critique from the reviewer. We have also removed the statement “rather than creating free-standing material architecture” in the revised manuscript.

Reviewer 3: *3. Ray-tracing vs 2D FE simulation*

One of the main claims in this manuscript is the much more efficient modelling, where finite element models of the mechanical vibration is only used for eigenmode analysis of individual unit cells and calculation of the dispersion relation/band structure. The long range model uses ray tracing. However, the authors do not fully describe the limits of the ray tracing model. By its nature, ray tracing is a high-frequency/short-wavelength approximation instead of the full wave equation(s), here further limited to out-of-plane bending waves. It neglects any wave coupling or interaction with the structure. The authors need to justify this approach by at least mentioning the relevant wave speeds/wavelengths for their materials and designs. The dispersion relation in Fig. 1 is insufficiently detailed to infer this information. While the experimental results appear to validate the ray-tracing, a comparison with a FEM wave equation result, maybe for a single tile, would be more convincing.

Response: The ray tracing model is indeed an approximation that has limitations, and we agree that the nature of this approximation merits deeper discussion. Specifically, our formulation of ray tracing in the metamaterial setting is based on the assumption that there is a separation of scales between the wavelength (the short length scale) and the length scale associated with spatial grading (the long length scale); see [23] for the derivation, which follows classical ray tracing derivations assuming the length scale of smooth variation of material properties is significantly longer than wavelengths of interest. Thus, sufficiently aggressive spatial grading with respect to wavelength may enter a regime not captured by ray theory, which would lead to both error in the ray solutions and its failure to capture physical phenomena like mode coupling.

To ensure the validity of ray theory, we rely on numerical/experimental validation. Of course, rigorous theoretical validity conditions would be ideal. While the above qualitative validity condition (grading length scale must be much larger than wavelength) provides rough guidelines, the development of rigorous quantitative validity conditions is an open problem. One approach for developing quantitative validity conditions would be to evaluate the discarded terms in the asymptotic expansion from which ray theory is derived. This has been explored, for example, in simpler settings such as an isotropic elastic continuum with constant gradient properties (see Ref. [S6] in the Supplementary Material); if such an analysis is tractable in our setting of ray theory in dispersive media with anisotropic microstructure and complex grading profiles, it is highly non-trivial and out of scope of this work.

We have added a discussion about ray theory validity to Supplementary Material Section 1.2. Additionally, as recommended by the reviewer, we have listed quantitative values for for wavelengths and wave speeds of the designs in Supplementary Material Sections 1.3.1 and 1.3.2. However, we caution that interpretation of these numbers toward ray theory validity is not straightforward, which is why we adopt numerical/experimental validation. Overall, we adopt the approach of using ray theory for design (due to its efficiency and differentiability), followed by transient numerical and experimental validation to confirm that the ray solution of the final design is in fact representative.

Finally, we note that transient dynamic finite element simulations are presented in Supplementary Material Section 2.2 both for assembled design 1 (Fig. S3) and assembled design 2 (Fig. S4), which show that the FE simulation closely follows the predicted ray trajectories. We feel that this is both more impactful and easier to interpret than an analogous simulation of a single tile—our reasoning is that in the assembled design the wave propagates in a closed loop so the solution can be interpreted with minimal effects of boundary conditions.

Reviewer 3: *More importantly, the authors need to put the higher efficiency of their modeling approach into quantitative values. Time, memory, CPU/GPU cycles or similar measures should be given at least in the supplementary material. A computational resources comparison of full wave equation model vs ray-tracing for maybe a single tile would elevate the claims. This may also include computational quantification of the inverse design procedure.*

Response: We agree that quantifying the efficiency is important for emphasizing the value of using a ray tracing model. A discussion has been added to the Supplementary Material Section 2.3, highlighting the numerical efficiency of ray tracing versus transient finite element simulations of the figure-eight example. To summarize this newly added discussion, it takes less than a second to trace one ray (~ 5.2 seconds to trace 100 rays to capture wave propagation through Tile 1 when running in parallel), while the parallelized transient dynamic finite element solution takes ~ 5 hours.

Additionally, we emphasize two additional key advantages of using ray tracing that make it suitable for inverse problems:

- Ray tracing quantities are differentiable. The (semi-)analytical derivative of a cost function based on ray quantities (specifically the position/wave vector at a give point on a ray) is derived in Eq. (S11) of the Supplementary Material. This is crucial for enabling gradient-based optimization, since finite differences are not needed for gradient computation, which are computationally expensive when the number of design variables is large and often introduce error.
- Ray tracing enables partial solutions to be computed. That is, we have the freedom to choose how many/which rays are traced, so it is possible to only compute the solution in a region of interest. In contrast, transient finite element simulations require computation of a complete full-field solution.

These additional points of discussion have been added to the Supplementary Material Section 2.3 to further elevate the value of using ray tracing to enable inverse design problems.

Reviewer 3: *4. Other medium/minor issues:*

All tile examples show only 90° turns. Are other angles not possible? If not, why?

Response: It is certainly possible to consider alternative objective functions in the optimal design problem to, for example, produce solutions with other than 90° turns. This corresponds to changing the exit wave vector $\hat{\mathbf{k}}_r$ in the cost function of Supplementary Material Eq. (S3), such that the corresponding group velocity at a ray's exit points in the desired direction. However, we choose examples with 90° turns to enable a straightforward modular design process where square tiles can be placed next to each other since rays are perpendicular to the tile edge. Alternative designs that tessellate the 2D domain are equally possible (such as, e.g., triangular or hexagonal tiles). We have added a discussion in Supplementary Material Section 1.3 to highlight this point. However, we feel that examples exploring the large design space of potential tile design and assembly concepts are beyond the scope of this work and require significant new computations (and also experiments for validation).

Reviewer 3:

Fig. 2: Font size becoming too small and hard to read. In general, font in figures should not be much smaller than in caption.

Response: Thank you for pointing this out. We have updated Fig. 2 to have a larger font size.

Reviewer 3:

Fabrication: "thin aluminum transducers (~ 30 nm thick)" are mentioned. Aluminum is not an active material and calling it a transducer like a piezoelectric material is strange. It appears to play the role of mirror/reflective surface for the acousto-optical transduction for excitation and detection.

Response: We thank the reviewer for pointing this out. The aluminum film serves the transduction of the optical to the mechanical pulse. We have modified this statement in the revised manuscript.

Reviewer 3:

Fig. 3: The photo in d) has a scale bar but no mention of the scale in the caption.

Response: A scale bar has been added to the caption.

Reviewer 3:

Fig. 4: Font size too small and hard to read.

Response: We have increased the font size in the revised figure.

Reviewer 3:

Fig. 4: order of a),c),b),d) is confusing.

Response: The order has been modified.

Reviewer 3:

Methods section: Several aspects are repeated from previous text, esp. doubling information regarding the microfabrication p. 6-7 and 9-10.

Response: The sections on microfabrication have been re-arranged in the revised manuscript to reduce the overlap.

Reviewer 3: *I have checked availability and type of code/files. Most calculation and data processing appears to use MATLAB but I have not reviewed the MATLAB code. Dispersion relations calculated from the finite element wave equation/eigenmode model are only provided as results, not the models/FEM codes. Many figure files are provided. However, experimental data is not sorted well and does not appear to include any file description or file naming scheme.*

Response: We thank the reviewer for carefully checking the code and experimental data.

Regarding the code, we note that we have included the inverse design codes, which encompasses all novel computational contributions of this work. The finite element-based dispersion relation computation and transient dynamic simulations are standard, well-known numerical procedures (e.g., in Refs. [2, 16]), and the underlying FE code we used, ae108, is open source – [S10] of the Supplementary Material references. Thus, we feel it is appropriate to keep the supplementary codes accompanying this manuscript concise and focused on our new contributions.

Regarding the experimental data, we have re-named the files and added a readme.txt file in the master directory, with short descriptions of the directory structure and data available in sub-directories. This new dataset has been uploaded to the ETH Zurich data repository under the new DOI given in the manuscript's data availability section.

References

- [1] Brillouin, L. Wave propagation in periodic structures. *McGraw-Hill* **2** (1946).
- [2] Hussein, M. I., Leamy, M. J. & Ruzzene, M. Dynamics of phononic materials and structures: Historical origins, recent progress, and future outlook. *Applied Mechanics Reviews* **66**, 040802 (2014).
- [3] Brûlé, S., Javelaud, E., Enoch, S. & Guenneau, S. Experiments on seismic metamaterials: molding surface waves. *Physical Review Letters* **112**, 133901 (2014).
- [4] Manushyna, D. *et al.* Application of vibroacoustic metamaterials for structural vibration reduction in space structures. *Mechanics Research Communications* **129**, 104090 (2023).
- [5] Dubček, T. *et al.* In-sensor passive speech classification with phononic metamaterials. *Advanced Functional Materials* **34**, 2311877 (2024).
- [6] Chen, Z., Guo, B., Yang, Y. & Cheng, C. Metamaterials-based enhanced energy harvesting: A review. *Physica B: Condensed Matter* **438**, 1–8 (2014).
- [7] Ma, J., Xi, X. & Sun, X. Experimental demonstration of dual-band nano-electromechanical valley-hall topological metamaterials. *Advanced Materials* **33**, 2006521 (2021).

- [8] Bigoni, D., Guenneau, S., Movchan, A. B. & Brun, M. Elastic metamaterials with inertial locally resonant structures: Application to lensing and localization. *Physical Review B—Condensed Matter and Materials Physics* **87**, 174303 (2013).
- [9] Jin, Y., Djafari-Rouhani, B. & Torrent, D. Gradient index phononic crystals and metamaterials. *Nanophotonics* **8**, 685–701 (2019).
- [10] De Ponti, J. M. *et al.* *Graded elastic metamaterials for energy harvesting* (Springer, 2021).
- [11] Maspero, F. *et al.* Phononic graded meta-mems for elastic wave amplification and filtering. *Journal of Microelectromechanical Systems* (2023).
- [12] Tol, S., Degertekin, F. L. & Erturk, A. Phononic crystal Luneburg lens for omnidirectional elastic wave focusing and energy harvesting. *Applied Physics Letters* **111** (2017).
- [13] Zhu, J. *et al.* Acoustic rainbow trapping. *Scientific Reports* **3**, 1728 (2013).
- [14] Meng, H. *et al.* 3d rainbow phononic crystals for extended vibration attenuation bands. *Scientific Reports* **10** (2020). URL <http://dx.doi.org/10.1038/s41598-020-75977-8>.
- [15] Dorn, C. & Kochmann, D. M. Conformally graded metamaterials for elastic wave guidance. *Extreme Mechanics Letters* **65**, 102091 (2023).
- [16] Trainiti, G., Rimoli, J. J. & Ruzzene, M. Wave propagation in undulated structural lattices. *International Journal of Solids and Structures* **97**, 431–444 (2016).
- [17] Lin, S.-C. S., Huang, T. J., Sun, J.-H. & Wu, T.-T. Gradient-index phononic crystals. *Physical Review B—Condensed Matter and Materials Physics* **79**, 094302 (2009).
- [18] Chen, H. & Chan, C. T. Acoustic cloaking and transformation acoustics. *Journal of Physics D: Applied Physics* **43**, 113001 (2010).
- [19] Nassar, H., Chen, Y. & Huang, G. Polar metamaterials: a new outlook on resonance for cloaking applications. *Physical Review Letters* **124**, 084301 (2020).
- [20] Nolde, E., Craster, R. & Kaplunov, J. High frequency homogenization for structural mechanics. *Journal of the Mechanics and Physics of Solids* **59**, 651–671 (2011).
- [21] Ramirez, L. A. P., Erel-Demore, F., Rizzi, G., Voss, J. & Madeo, A. Effective surface forces and non-coherent interfaces within the reduced relaxed micromorphic modeling of finite-size mechanical metamaterials. *Journal of the Mechanics and Physics of Solids* **186**, 105558 (2024).
- [22] Demore, F., Rizzi, G., Collet, M., Neff, P. & Madeo, A. Unfolding engineering metamaterials design: Relaxed micromorphic modeling of large-scale acoustic meta-structures. *Journal of the Mechanics and Physics of Solids* **168**, 104995 (2022).
- [23] Dorn, C. & Kochmann, D. M. Ray theory for elastic wave propagation in graded metamaterials. *Journal of the Mechanics and Physics of Solids* **168**, 105049 (2022).
- [24] Dorn, C. & Kochmann, D. M. Inverse design of graded phononic materials via ray tracing. *Journal of Applied Physics* **134** (2023).
- [25] Telgen, B. *et al.* Rainbow trapping of out-of-plane mechanical waves in spatially variant beam lattices. *Journal of the Mechanics and Physics of Solids* **191**, 105762 (2024).
- [26] Schaeffer, M., Trainiti, G. & Ruzzene, M. Optical measurement of in-plane waves in mechanical metamaterials through digital image correlation. *Scientific Reports* **7** (2017). URL <http://dx.doi.org/10.1038/srep42437>.
- [27] Cha, J., Kim, K. W. & Daraio, C. Experimental realization of on-chip topological nanoelectromechanical metamaterials. *Nature* **564**, 229–233 (2018). URL <http://dx.doi.org/10.1038/s41586-018-0764-0>.

- [28] Shaikeea, A., Cui, H., O'Masta, M., Zheng, X. R. & Deshpande, V. S. The toughness of mechanical metamaterials. *Nature Materials* **21**, 297–304 (2022).
- [29] Wormser, M., Wein, F., Stingl, M. & Körner, C. Design and additive manufacturing of 3d phononic band gap structures based on gradient based optimization. *Materials* **10**, 1125 (2017). URL <http://dx.doi.org/10.3390/ma10101125>.
- [30] Kruisová, A. *et al.* Ultrasonic bandgaps in 3d-printed periodic ceramic microlattices. *Ultrasonics* **82**, 91–100 (2018). URL <http://dx.doi.org/10.1016/j.ultras.2017.07.017>.
- [31] Rice, H., Kennedy, J., Göransson, P., Dowling, L. & Trimble, D. Design of a kelvin cell acoustic metamaterial. *Journal of Sound and Vibration* **472**, 115167 (2020). URL <http://dx.doi.org/10.1016/j.jsv.2019.115167>.
- [32] Bauer, J. *et al.* Nanolattices: an emerging class of mechanical metamaterials. *Advanced Materials* **29**, 1701850 (2017).
- [33] Saccone, M. A., Gallivan, R. A., Narita, K., Yee, D. W. & Greer, J. R. Additive manufacturing of micro-architected metals via hydrogel infusion. *Nature* **612**, 685–690 (2022). URL <http://dx.doi.org/10.1038/s41586-022-05433-2>.
- [34] Kotz, F. *et al.* Two-photon polymerization of nanocomposites for the fabrication of transparent fused silica glass microstructures. *Advanced Materials* **33** (2021). URL <http://dx.doi.org/10.1002/adma.202006341>.
- [35] Skliutas, E. *et al.* Multiphoton 3d lithography. *Nature Reviews Methods Primers* **5** (2025). URL <http://dx.doi.org/10.1038/s43586-025-00386-y>.
- [36] Jonušauskas, L. *et al.* Mesoscale laser 3d printing. *Optics Express* **27**, 15205 (2019). URL <http://dx.doi.org/10.1364/OE.27.015205>.
- [37] Lee, J.-H., Singer, J. P. & Thomas, E. L. Micro-/nanostructured mechanical metamaterials. *Advanced Materials* **24**, 4782–4810 (2012).
- [38] Bückmann, T., Thiel, M., Kadic, M., Schittny, R. & Wegener, M. An elasto-mechanical unfeelability cloak made of pentamode metamaterials. *Nature Communications* **5**, 4130 (2014).
- [39] Meza, L. R. *et al.* Reexamining the mechanical property space of three-dimensional lattice architectures. *Acta Materialia* **140**, 424–432 (2017).
- [40] Harinarayana, V. & Shin, Y. Two-photon lithography for three-dimensional fabrication in micro/nanoscale regime: A comprehensive review. *Optics & Laser Technology* **142**, 107180 (2021).
- [41] Krödel, S. & Daraio, C. Microlattice metamaterials for tailoring ultrasonic transmission with elastoacoustic hybridization. *Physical Review Applied* **6**, 064005 (2016).
- [42] Kiefer, P. *et al.* A multi-photon (7 × 7)-focus 3d laser printer based on a 3d-printed diffractive optical element and a 3d-printed multi-lens array. *Light: Advanced Manufacturing* **4**, 28–41 (2024).
- [43] Zhang, L. *et al.* High-throughput two-photon 3d printing enabled by holographic multi-foci high-speed scanning. *Nano Letters* **24**, 2671–2679 (2024). URL <http://dx.doi.org/10.1021/acs.nanolett.4c00505>.
- [44] Li, Z. *et al.* Inverse design enables large-scale high-performance meta-optics reshaping virtual reality. *Nature Communications* **13**, 1–11 (2022).
- [45] Li, Z., Pestourie, R., Lin, Z., Johnson, S. G. & Capasso, F. Empowering metasurfaces with inverse design: principles and applications. *ACS Photonics* **9**, 2178–2192 (2022).
- [46] Moore, D. T. Gradient-index optics: a review. *Applied Optics* **19**, 1035–1038 (1980).

- [47] De Ponti, J. M. *et al.* Selective mode conversion and rainbow trapping via graded elastic waveguides. *Physical Review Applied* **16** (2021). URL <http://dx.doi.org/10.1103/PhysRevApplied.16.034028>.
- [48] Cervený, V. *Seismic ray theory* (Cambridge University Press Cambridge, 2001).
- [49] Born, M. & Wolf, E. *Principles of optics: Electromagnetic theory of propagation, interference and diffraction of light* (Elsevier, 2013).
- [50] Zelhofer, A. J. & Kochmann, D. M. On acoustic wave beaming in two-dimensional structural lattices. *International Journal of Solids and Structures* **115**, 248–269 (2017).
- [51] Kannan, V., Dorn, C., Drechsler, U. & Kochmann, D. M. Microscale architected materials for elastic wave guiding: Fabrication and dynamic characterization across length and time scales (2025). URL <https://arxiv.org/abs/2507.01757>. 2507.01757.
- [52] Sledzinska, M., Graczykowski, B., Alzina, F., Santiso Lopez, J. & Sotomayor Torres, C. Fabrication of phononic crystals on free-standing silicon membranes. *Microelectronic Engineering* **149**, 41–45 (2016). URL <http://dx.doi.org/10.1016/j.mee.2015.09.004>.
- [53] Olsson III, R. H., El-Kady, I. F., Su, M. F., Tuck, M. R. & Fleming, J. G. Microfabricated vhf acoustic crystals and waveguides. *Sensors and Actuators A: Physical* **145**, 87–93 (2008).
- [54] Hopkins, P. E. *et al.* Reduction in the thermal conductivity of single crystalline silicon by phononic crystal patterning. *Nano Letters* **11**, 107–112 (2010). URL <http://dx.doi.org/10.1021/nl102918q>.
- [55] Mohammadi, S., Eftekhari, A. A., Khelif, A., Hunt, W. D. & Adibi, A. Evidence of large high frequency complete phononic band gaps in silicon phononic crystal plates. *Applied Physics Letters* **92** (2008). URL <http://dx.doi.org/10.1063/1.2939097>.
- [56] Mustafazade, A. *et al.* A vibrating beam mems accelerometer for gravity and seismic measurements. *Scientific Reports* **10** (2020). URL <http://dx.doi.org/10.1038/s41598-020-67046-x>.
- [57] Jenni, L. V., Kumar, L. & Hierold, C. Hybrid lithography based fabrication of 3d patterns by deep reactive ion etching. *Microelectronic Engineering* **209**, 10–15 (2019). URL <http://dx.doi.org/10.1016/j.mee.2019.02.009>.
- [58] Loh, O., Vaziri, A. & Espinosa, H. D. The potential of MEMS for advancing experiments and modeling in cell mechanics. *Experimental Mechanics* **49**, 105–124 (2007). URL <http://dx.doi.org/10.1007/s11340-007-9099-8>.
- [59] Sharpe, W. N. *et al.* Strain measurements of silicon dioxide microspecimens by digital imaging processing. *Experimental Mechanics* **47**, 649–658 (2007). URL <http://dx.doi.org/10.1007/s11340-006-9010-z>.
- [60] Chasiotis, I. *Experimental Mechanics of MEMS and Thin Films*, 3–37 (Springer Netherlands, 2004). URL http://dx.doi.org/10.1007/978-94-007-1013-9_1.
- [61] Thelen, M., Bochud, N., Brinker, M., Prada, C. & Huber, P. Laser-excited elastic guided waves reveal the complex mechanics of nanoporous silicon. *Nature Communications* **12**, 3597 (2021).
- [62] Zega, V. *et al.* Microstructured phononic crystal isolates from ultrasonic mechanical vibrations. *Applied Sciences* **12**, 2499 (2022).
- [63] Hu, G., Tang, L., Liang, J., Lan, C. & Das, R. Acoustic-elastic metamaterials and phononic crystals for energy harvesting: A review. *Smart Materials and Structures* **30**, 085025 (2021).
- [64] Krushynska, A. O. *et al.* Emerging topics in nanophononics and elastic, acoustic, and mechanical metamaterials: an overview. *Nanophotonics* **12**, 659–686 (2023). URL <http://dx.doi.org/10.1515/nanoph-2022-0671>.
- [65] Charara, M., Kujala, Z., Lee, S. & Gonella, S. Spatially selective drop-motion programming using metamaterials. *Proceedings of the Royal Society A* **481**, 20240429 (2025).

Second Revision of NCOMMS-25-56068-T: “**Graded phononic metamaterials: scalable design meets scalable microfabrication**” by Charles Dorn, Vignesh Kannan, Ute Drechsler, and Dennis M. Kochmann

We thank the reviewers for their thoughtful and constructive comments. We have taken them as basis for the revision of our manuscript.

In the following, please allow us to respond to the reviewer’s questions and comments – explaining the changes made in our revised manuscript (which are highlighted in red in the attached manuscript). We refer to the reference numbering of the second revision of the manuscript, which is copied below. Please note that we have also taken the chance to clarify the labeling in the schematic of Fig. 3(e).

Additionally, please note that the affiliation of author Vignesh Kannan has been revised from “Laboratoire de Mécanique des Solides, École Polytechnique, Palaiseau, 91128, France” to “Laboratoire de Mécanique des Solides, CNRS, École Polytechnique, Institut Polytechnique de Paris, Palaiseau, 91128, France”. This does not reflect a change in the author’s institutional affiliation but is merely a change in the institution’s recommended nomenclature to include parent organizations as per the French academic system.

Reviewer 1:

The comments have been addressed satisfactorily. I thus recommend publication of the paper in its present form.

Response: Thank you for the positive feedback.

Reviewer 2: *While the authors have made substantial replies in the rebuttal letter, I do not feel substantive improvements were made to the manuscript regarding modifying their arguments to accurately place their contribution in the context of existing and prior capabilities. Key references that would conflict with their arguments of novelty are still missing.*

I do not believe this is appropriate to be published anywhere in its current form. Even if their description of the context of their contribution is improved, this work does not have sufficient novelty for Nature Communications. Scientific Reports or NJP Metamaterials could be alternate suitable venues.

Response: We strongly disagree with the reviewer’s assessment, and we are glad that Reviewers 1 and 3 also do not agree. Before we respond in detail to each point raised by Reviewer 2, we would like to make a general comment.

Reviewer 2 does not criticize the appropriateness of our methods nor the validity of our results, but they merely make a lack of “*sufficient novelty*” the basis of their negative assessment of our manuscript. We previously agreed with the reviewer that our manuscript leverages well-established techniques of microfabrication and of wave modeling, so the novelty does not lie in the general techniques being used. However, we combine and extend existing experimental and numerical techniques to design, fabricate, and characterize a *smoothly spatially-graded phononic metamaterial* with the largest number of unit cells ever reported in the literature to the best of our knowledge (and over an extremely wide frequency range that is unusual for phononic metamaterials). Spatial gradings are of significant interest in the context of wave physics and metamaterials (from transformation elastodynamics to dispersive wave guidance), yet to date all approaches have relied on significantly smaller systems (in terms of the number of unit cells) and/or on considerably simpler designs (e.g., periodic structures with or without sharp interfaces). The number of unit cells matters, as theories based on homogenization require a separation of scales. From a modeling perspective, having a playground to experimentally explore smoothly spatially-graded designs is key to exploring the opportunities and limitations of dynamic homogenization – which is exactly what our approach offers (while also exhibiting minimal damping, minimal geometric defects, and excellent scalability). In addition, our manuscript introduces a new tiling design strategy for metamaterials, a new spatially graded design parameterization,

a new optimization formulation, and a new multiscale setting of micro- (beam and unit cell), meso- (tile) and macro-scales (boundary value problem) – to mention but a few of the novel aspects of our manuscript. As we elaborate in our responses below, several (very recent) review papers in the metamaterials literature [7, 8, 9] have highlighted the need for scalable design and fabrication approaches, which is what we are addressing. Reviewer 2 further criticizes that we do not present practical applications, yet this disregards the fundamental scientific relevance: the presented setup can be used to explore many interesting wave physics questions, beyond technological devices. For all of these reasons, we do believe that our manuscript warrants consideration for publication in Nature Communications.

In the following, we will respond to each point of criticism raised by the reviewer.

Reviewer 2: *Here is just one other, again uncited, example of using laser ultrasonics to measure phononic responses of patterned plates [Zhao, Jinfeng, et al. “On-chip valley phononic crystal plates with graded topological interface.” International Journal of Mechanical Sciences 227 (2022): 107460.]. In that reference they study 1.7-1.9 MHz waves in a patterned silicon plate with patterned area stretching over 31x37 cm within a 4 inch diameter wafer.*

Response: We, of course, acknowledge that there is a rich body of literature on phononic waveguides as well as on micro- and nanoscale structures fabricated by MEMS/NEMS technology. We appreciate the reviewer pointing out references that are related to our manuscript, including this one (there is likely a long list of references that are tangentially related but cannot be included for practical reasons). As also mentioned before, we agree with the reviewer that “*using laser ultrasonics to measure phononic responses of patterned plates*” is not a new invention, as stated in our previous response letter (where we cited [30]). However, the ability to develop microarchitecture through predictive computational design for arbitrary wave guiding is still a challenge. MEMS/NEMS technology can fabricate millions of small-scale unit cells on a wafer, but computational design has been lacking the tools to efficiently optimize for wave motion in such complex, spatially graded architectures.

With reference to the suggested paper (Zhao et al., 2022), the authors there engineered interfaces with a specific periodic lattice geometry, which is much simpler than the spatially varying designs presented here. That manuscript focuses on the engineering of interfaces, which is an important albeit considerably smaller subset of microarchitecture designs for wave motion. Furthermore, the excitation of waves in that paper is achieved by PZT plates (hence, *not* optical pump-probe laser ultrasonics), which exhibit limited control in frequency range and spatial distribution for excitation. Completely optical pump-probe setups (such as our system) deliver significantly better modularity and control of excitation (see, e.g., [50, 51]). Laser ultrasonics is a relatively vast topic applied across various material systems and length scales, hence a fully detailed literature review was not within the scope of this manuscript.

In light of this critique from the reviewer, these citations (i.e., [50, 51]) have been added to the revised manuscript.

Reviewer 2: *The authors’ focus in the rebuttal letter on subtle difference from articles brought to their attention by the Reviewers misses the point. Reviewers should not have to do the authors’ literature review for them.*

Below, I will also respond to the subset of their replies that I feel warrant particular response.

Author: “the main focus of this manuscript is the application of our experimental-computational framework to design of new elastic wave guides across micrometer to millimeter length scales, like the figure-8 case discussed.”

I do not understand what the experimental part is in “experimental-computational framework”. It seems the design is purely computational, then there is experimental validation. Neither the fabrication method nor the characterization method is substantially new.

Response: To us, this is a matter of semantics (and we refer to the comments made above about sufficient vs. insufficient novelty). When speaking of a new “experimental-computational framework”, this does not necessarily imply that (a) all experimental or computational details are new and (b) that there is two-way

feedback between experiments and simulations. We introduce a new computational approach to designing large spatially graded waveguides, which would have little merit without an experimental route to fabricate such structures (and which additionally benefits from an experimental validation). At the same time, MEMS/NEMS technology to make such wafer-scale structures has been available for a long time – as the reviewer rightly points out. Yet, it has not been used to produce the type of graded waveguides reported here, among others exactly because appropriate design tools were previously lacking. In our revised manuscript, we do not claim that the wafer fabrication route is novel nor that we are the first to use interferometry to characterize wave motion. Yet, we are the first to combine our novel design and optimization tools with a suitable fabrication approach that can produce hundreds of thousands of spatially graded unit cells in a single, smoothly graded structure and characterize the wave motion in situ for a successful validation. This is what we refer to as our “experimental-computational framework”.

Reviewer 2: *In addition, I am unclear from the manuscript as to why someone would want to make these large gradient based structures, when the same thing can be done in a much smaller footprint. For instance, conventional (not even topological or phononic-crystal-based) elastic waveguides have been around for many decades. A broadband, figure 8 waveguide could be easily done in this conventional context via design of impedance mismatch and total internal reflection.*

Response: We would like to re-emphasize that, as the reviewer points out, there are many wave guiding methodologies, each with their benefits and limitations. The aim of this research is not to produce a superior figure-eight waveguide with respect to all other potential wave guiding approaches. The figure-eight example provides a platform for numerical and experimental validation, as we explore the design space of spatially graded architectures spanning multiple length scales. This expansion of the design space, specifically to include gradings on multiple length scales, may unlock new capabilities not possible with simpler designs (the design of Fig. S2 as one example). Using reflections (as a consequence of impedance mismatch, as pointed out by the reviewer) is a different physical mechanism that can be exploited. Using smooth gradings instead has benefits such as slow variations in other physical properties (e.g., mechanical stiffness and strength and therefore lower stress concentrations). Interfaces may also excite other Bloch branches, be frequency-selective, or lead to parasitic scattering. Our aim is to demonstrate an approach that enables smooth spatial gradings to accommodate new opportunities for wave guidance.

Reviewer 2: *Author: In general, we stress that it was not our intention to claim that we have invented a new technique in microfabrication. We apologize if we inadvertently created that impression. Studies that have breached the nanoscale (like the one that the reviewer correctly pointed out) have achieved free-standing structures of hundreds of micrometers. In certain applications of mechanical wave guiding, however, it may be critical to guide waves over much longer distances; this is non-trivial to achieve.*

What applications, specifically, are these critical for? This has still been left as nebulous in the manuscript.

Response: We acknowledge that, as the reviewer correctly points out, this research is not tied to a specific application. This is curiosity-driven research in pursuit of exploring the design space of complex, multiscale spatial gradings. We strongly believe there is scientific merit in exploring phononic metamaterial dynamics within this largely unexplored design space. While our interest is of scientific nature and we do not develop a specific application, we are confident that the tools and knowledge presented in this manuscript will capture the interest of the metamaterials and broader wave physics communities and may lead to novel applications and devices. Our broad intent is that expanding the metamaterial design space expands the achievable functionalities. This has been evident in the past, for example, when the expansion of the design space from periodic architectures to simple spatial gradings increased functionalities (e.g., metamaterial Luneburg lenses) driving interest in applications such as energy harvesting and signal processing. Our work further expands the grading design space to complex, multiscale architectures on a fundamental level. The precise future application domains that benefit from our fundamental scientific contributions are yet to be determined. Complex wave manipulation functionalities over long distances with respect to the unit cell scale and wavelength is made possible. Moreover, the fact that we can guide and characterize waves in spatially graded structures with hundreds of thousands to millions of unit cells presents a new, versatile, and powerful experimental platform for research in wave motion (going far beyond the specific unit cell design and optimization objectives presented here).

Reviewer 2: *Author: We have used existing microfabrication techniques to create elastic wave-guiding metamaterials across length scales spanning micrometers to millimeters – a niche range that we refer to as a “meso” scale. Our manuscript presents in detail the advantages and challenges of accessing this “intermediate” range.*

What advantages and where are they stated? Do you mean “This admits controlling wave motion in the tens of kHz to MHz regime across centimeters – a regime of long-standing interest to the field”. If so: i) this is still nebulous; ii) Someone controlling kHz to MHz waves would just use larger structures there is no need to use few micron beams (see paper cited above).

Response: We appreciate that reviewer acknowledging that “controlling wave motion in the tens of kHz to MHz regime across centimeters” is novel and at present possible only with significantly “larger structures”. First, we again stress that scientific novelty without an immediate technological application in our view does warrant publication in a scientific journal. Second, the small-scale design presented in our work admits easy integration into devices with limited operating volume. For example, on-chip energy harvesters or mechanical signal processors cannot include significantly larger systems. Finally, to “just use larger structures” would limit the range of frequencies and length scales within which wave guiding is possible. Therefore, the control of architecture across length scales is essential to controlling a “broadband” response.

Reviewer 2: *Author: For example, if we could fabricate nanoscale structures on a free-standing membrane spanning tens of millimeters (like ours), this would be a momentous step forward in scalable wave-guiding technology, e.g., for controlled design of the “long wavelength” dispersion response of mechanical waves in signal processing.*

How is this “momentous” exactly? What would this enable in terms of new phenomenology or functionality? Why would one just not use larger unit cells (see paper cited above)?

Response: With regards to enabling of new functionality, we refer to our previous response. Using larger unit cells limits the length scales over which wave guiding can be achieved (and requires a larger footprint). Working on the simple assumption that the unit cell size imposes the smallest length scale over which a wave path in physical space can be guided, achieving the physically smallest possible unit cell size while controlling the length scale over which the unit cell architecture is varied takes us one step closer to achieving independent control of elastic waves in the real and Fourier spaces.

An extreme example would be achieving billions of nano- to micro-scale unit cells over an 8-inch free-standing film (as opposed to our 4-inch ones). With the right computational design and experimental tools, we can guide kHz to GHz waves across distances over a hundred millimeters, with wave-paths achieving curvatures on the hundreds-of-nanometer length scales (while our paper does not reach billions of unit cells and GHz frequencies, we take a significant step toward scalable broadband wave control). While still a fundamental study, the applications in high-throughput cell sorting, materials characterization, broadband signal processing, encoding and decoding, to name a few, could well be imagined. This requires true multi-scale capabilities in fabrication, and more importantly computational design, that cannot be achieved just through simply changing the size of unit cells.

Reviewer 2: *Author: Our “meso” scales also allow measurements of the full spatio-temporally resolved wave propagation with sufficiently high resolution. Such high-dimensional experimental data are crucial to validate computational models, verify and realize complex waveguide designs obtained from optimization schemes, as demonstrated in this study.*

I do not understand this comment. Laser ultrasonic characterization, including full field imaging has been around for decades (e.g. [Tachizaki, Takehiro, et al. “Scanning ultrafast Sagnac interferometry for imaging two-dimensional surface wave propagation.” Review of Scientific Instruments 77.4 (2006)], or the paper cited above)

Response: This may be a matter of semantics, but we do not see where the reviewer’s statement invalidates ours. We agree that laser ultrasonic characterization has existed for a long time. We previously referred to [30], where laser ultrasonic characterization was used. Laser ultrasonics is a relatively vast topic applied across various material systems and length scales, hence a fully detailed literature review was not within

the scope of this manuscript, though some of those references have been added to the revised manuscript [49, 50, 51]. As pointed out in previous responses, a key novelty of our work lies in the design and fabrication of smoothly spatially-graded structures containing hundreds of thousands of unit cells, which we achieve via microfabrication. This in turn requires a suitable characterization technique for measurements of the full spatio-temporally resolved wave propagation. It is for this purpose that we used laser ultrasonic characterization.

Reviewer 2: *Author: Prior works in the MEMS and NEMS communities (which are clearly relevant here from a microfabrication perspective), to our knowledge have not explored large, free-standing microstructured thin films (spanning micrometers to millimeters)—owing, among others, to the prevalent interest in resonator technology and surface acoustic waves (SAWs), which do not require structures to be free-standing across a large spatial domain. However, such large, free-standing structures naturally enable the propagation of more complex wave modes (e.g., Lamb waves).*

I do not understand this comment. Lamb waves are routinely considered in phononic crystal design – see, e.g., another, uncited, example [Hyun, Jaeyub, Wonjae Choi, and Miso Kim. “Gradient-index phononic crystals for highly dense flexural energy harvesting.” Applied Physics Letters 115.17 (2019)].

Response: Thank you for pointing out this mistake in our wording. We meant to say *Bloch waves* instead of *Lamb waves* in this statement.

The reviewer is correct that Lamb waves in continuum (or homogenized) free-standing plates have been well understood and utilized. Bloch waves, which interact with the microstructure above the homogenization limit, have been much less utilized – especially in the context of spatial grading. For example, the reference pointed out (Hyun et al., 2019) considers an effective medium approximation, restricting the applicability of their approach to the low-frequency homogenization limit. By leveraging ray tracing, our work captures the complete local dispersion relations thus capturing the Bloch modes above the homogenization limit.

Reviewer 2: *Author: As a side note, Sledzinska et al. (2016) focused on stabilizing ultra-thin films, using a polymer layer. While a novel technique, it is unclear how waves can propagate in that configuration over long distances due to damping—especially when aiming for lower frequencies, as discussed here. Discrepancies due to damping may affect quantitative comparisons of the full spatio-temporal evolution of waves through those wave guides.*

There are many suspended membrane studies that do not use polymer layers. Here is another, uncited, example [Graczykowski, Bartłomiej, et al. “Phonon dispersion in hypersonic two-dimensional phononic crystal membranes.” Physical Review B 91.7 (2015): 075414].

Response: As before, we agree with the reviewer that nanoscale device fabrication is a well-established field. Given this fact, it is impractical to cite all studies on nano-fabrication, especially since our targeted range of length and time scales, as well as our focus on *spatially graded* architectures, are very different from previous work in this vast field.

Reviewer 2: *Author: Finally, Sledzinska et al., (2016) and several others within this domain focused on improving the fabrication of nanoscale waveguides rather than spatial grading for pre-defined wave guiding capabilities. As has been evident from literature [41, 12, 14, 47], this is a non-trivial problem with recent progress at macroscopic scales (including our own prior work, e.g., [25, 15]). The present manuscript demonstrates significantly improved synergy between scalability in computation, fabrication, and measurement for more robust spatial grading strategies.*

I do not understand this comment. Most other papers just treat this “synergy” as experimental validation of a computationally designed metamaterial.

Response: We refer to our previous responses: this is a matter of semantics to us. By “synergy” we refer to the benefits of being able to simulate, optimize, fabricate, and test complex, spatially graded phononic structures covering three length scales (from unit cell to mesoscale to structure). Simply put, our use of the term “synergy” refers to the added benefits of combining experimental fabrication and characterization and design optimization. Microfabrication has long enabled small-scale structures, but suitable optimization

techniques for the graded designs presented here have been lacking. Ray tracing-based optimization of metamaterials was previously introduced (by us), but without the tiling strategy introduced here could not reach the relevant scales. Combining the two yields synergy, especially when complemented by laser ultrasonic characterization for validation. This is what we refer to by “synergy”.

Reviewer 2: *Author: The presented large-scale waveguides are made possible by the “tile design” approach, which is introduced here for the first time (and nontrivial due to, among others, requirements of matched dispersion relations at tile interfaces, treatment of boundaries, and the design targets required in such tiles). Brute-force optimization of the entire domain based on [32] would not be computationally feasible for the designs considered in this manuscript.*

There is no proof by the authors that this brute force optimization is not feasible, particularly given the used (and prior demonstrated in Ref. [24]) Ray tracing algorithm. Why is the tiling needed? There is no evidence.

Response: Thank you for pointing this out, and we agree that the advantage of a modular, tile-based design over brute-force optimization merits further discussion. We cannot, in absolute terms, prove the negative – that brute-force optimization on a large domain is not feasible with arbitrary computational resources. However, we can provide a strong argument that brute-force optimization becomes significantly less effective (to the point of being practically infeasible) for larger design domains and more complex cost functions.

First, as the design domain grows, brute-force optimization would see an increase in design variables. For example, design of a single 64×64 unit cell tile, which has one design parameter per unit cell, involves optimization over 3844 design variables (after removing design variables on the boundary). This optimization takes on the order of hours (depending on parallelization and specific computing hardware). Thus, each independently designed tile involves a 3844 variable design problem. Modular design by tile assembly introduces **no additional computation time** once individual tiles are designed, with **no limit on scalability** (e.g., Fig. S2 presents a 448×320 unit cell design). Alternatively, brute-force optimization scales poorly with increasing domain size. As an example, optimization of a 448×320 unit cell domain would require optimization over $\sim 143,000$ variables.

A second barrier to brute-force optimization is non-convexity. While the optimization of a single tile is non-convex, we demonstrate that it is within reach to find global optima (i.e., we found designs that send the cost function to zero, which is the case for Tiles 1 and 2 of the manuscript). In our experience, as the number of design variables grows and as the complexity of cost function increases, it is increasingly difficult to navigate the nonconvex design space, to the point that brute-force optimization becomes practically infeasible. However, we prefer to refrain from presenting a detailed study of failed brute-force optimization attempts.

A discussion clarifying the challenges of brute-force optimization and the advantages of modular design has been added to the *Modular Waveguide Design* section of the main text.

Reviewer 2: *Author: Ref. [24] exclusively optimized rays based on target locations (e.g., wave focusing). Here, we present for the first time how to optimize designs such that rays enter and exit perpendicular to the boundary (which is essential for the tiling to function). Ref. [24] only implements mass-spring network toy problems with analytical dispersion relations; the current manuscript’s demonstration of beam networks with numerically computed dispersion relations is a major improvement that is key for enabling experimental demonstration.*

These both seem like relatively incremental advances in the opinion of the reviewer.

Response: We kindly disagree. Of course, the judgment of a research contribution as transformative versus incremental is subjective. However, we argue that our manuscript takes a significant and nontrivial step to transform a theoretical framework (i.e., the authors’ previous work of [27] – ref. [24] of the previous revision) previously demonstrated only on a simple, non-realistic toy problem (a mass-spring network with analytical dispersion relations) to physically realistic system that can be implemented and rigorously validated experimentally. In addition, we make significant generalization to ref. [27], making a much richer and more general family of design objectives within reach. Considering these advances with respect to previous work, we feel that the innovation of this work compared to ref. [27] is substantial and will be of notable interest to the

metamaterials, mechanics, and wave physics communities.

Reviewer 2: *Author: Thank you for pointing out that this wording is not precise. We have updated the wording to “Modeling, design, and manufacturing efficiency remains a bottleneck for architectures spanning tens of thousands to millions of unit cells...” We point out that the Springer-Nature journal family does not appreciate precise novelty claims (“we ask authors to refrain from making priority or novelty claims and to remove qualitative evaluations of their own work”; “avoid phrases like ‘for the first time’ and ‘unprecedented’).*

The statements the authors make must still be accurate. In the Reviewer’s opinion, given the prior references and related work in other fields, this is not a bottleneck as the authors have claimed.

Response: We respectfully disagree with the reviewer, and we would like to point out that the challenge of scalability is frequently acknowledged in contemporary review papers that discuss the state of the art in phononic metamaterials. Examples of phononic metamaterial reviews from 2024-2025 commenting on the scalability bottleneck include the following:

- The authors of [7] state that “Advances in numerical modelling techniques will allow researchers to explore a vast design space efficiently. On the other hand, scalable production methods and assessing the long-term durability and stability of metamaterials in various environments are crucial considerations not only for metamaterials designed for acoustic absorption but also for any material or technology.”
- The authors of [8] state that “On the practical side, developing innovative fabrication techniques is essential for constructing the most challenging and advanced metamaterial structures. Current limitations in manufacturing, such as achieving the necessary precision and scalability, hinder the practical deployment of metamaterials in real-world devices.”
- The authors of [9] state that “Other notable challenges are, for example, the need to develop scalable manufacturing techniques.”

The authors of the above review papers collectively include a significant list of prominent contributors to the field of phononic metamaterials, indicating that there is notable consensus that scalability is an important bottleneck faced by current metamaterials research, and that there is significant interest in research efforts toward scalability (and we therefore feel that our manuscript will be of significant interest to the community).

Thus, we believe that our wording of the sentence pointed out by the reviewer accurately represents the consensus of much of the research community while avoiding absolute statements.

Reviewer 2: *Author: When referring to a “widely untapped design space”, we are specifically referring to spatially graded architectures with complicated spatial profiles spanning multiple length scales. As mentioned in paragraph 3 of the introduction, existing work on spatially graded phononic metamaterials has been restricted to simple grading profiles—such as grading along one dimension only, or radially symmetric patterns. Two-dimensional spatial gradings with complicated grading profiles spanning multiple length scales have not been previously considered. Scaling to large architectures with many unit cells creates the opportunity to explore gradings with more elaborate spatial structures. We are not tied to a particular application, as we present a fundamental framework for exploring what new capabilities are possible in a previously unexplored design space of complex, multiscale, 2D spatial gradings. However, we anticipate that expanding the design space and functionality of graded metamaterials will enhance applications that graded metamaterials (with simple grading profiles) have shown promise for already, such as energy harvesting [10] and mechanical/acoustic signal processing [11], in addition to potentially enabling new applications that could benefit from enhanced control over wave guiding, localization, and focusing, such as microfluidics [65].*

The authors still have not clearly stated how this would be beneficial over simpler solutions. They have not demonstrated how energy harvesting, acoustic signal processing, focusing, or microfluidics would be improved via this design, compared to simpler, existing approaches.

Response: The objective of this work is not to directly advance applications such as energy harvesting or acoustic signal processing. The objective of this work is to provide knowledge and tools to expand the metamaterial design space such that complex, multiscale spatially graded designs are within reach. We

argue that all expansions of the metamaterial design space thus far have proven fruitful. For example, simple gradings (e.g., to create Luneburg lenses) are more complex than periodic architectures; this added complexity increases metamaterial functionality to include wave focusing capabilities. While we concentrate on spatial gradings, there are many similar examples in other regions of the metamaterial design space where adding non-periodic complexity increases functionality (e.g., interfaces and defects). Thus, we are confident that the expansion of the metamaterial design space to multiscale spatial gradings presented in this work will inspire new metamaterial functionalities and applications. Due to the success of simple spatial gradings for applications such as energy harvesting and signal processing, we explicitly mention those as potential future application domains. We specifically anticipate that our results will provide a foundation for on-chip sensing and energy harvesting applications, where larger-scale metamaterial systems are not feasible. However, we do not claim to specifically advance device design in those application domains.

Reviewer 2: *Author: We acknowledge that many approaches to wave guiding have been explored. Topological metamaterials, such as the example of (Ma et al., 2021), are appealing in many cases, e.g., due to their robustness to defects. One potential drawback is that they are restricted to often narrow bandgaps in the bulk, and to discrete wave guiding angles aligned with lattice interfaces. Similarly, many other approaches to acoustic and phononic metamaterials, which leverage bandgap design, are limited to narrow frequency bands, unlike the results presented here. The introduction has been updated to reflect these points.*

Topological waveguides were just one example. There are many more examples of topologically trivial waveguides simply by virtue of effective elastic property mismatches, which do not have a bandwidth limitation.

Response: Following our previous responses, we again acknowledge that there are many approaches to wave guiding in metamaterials, each with certain benefits and limitations. Here, we were simply pointing out that the topological waveguide example mentioned by the reviewer is typically band-limited. Other approaches (i.e., different concepts for topologically trivial waveguides) have different benefits and limitations. We do not believe this conflicts with our contributions, which explore multiscale spatial gradings spanning many unit cells – a setting that has not previously been explored.

Reviewer 2: *Author: “Thank you for pointing out the lack of precision in this statement. We have modified these sentences to the following: “... Going from tens to hundreds of thousands of unit cells at the meso-scale requires the adaptation of specialized microfabrication techniques that have rarely been exploited for free-standing elastic wave guides [27] – especially spanning tens of micrometers to millimeters.”*

“Rarely” does not contribute to the scientific novelty of the work.

Response: This is again a matter of semantics. Within the rich literature of phononic metamaterials and MEMS/NEMS devices we cannot cite every possible publication in this sentence. Those “rare” cases we referred to did not investigate spatially graded, free-standing elastic waveguides across the current length and frequency ranges. Thus, we believe that this does constitute novelty.

Reviewer 2: *Author: Thank you for pointing out this incorrect statement. We were referring to conventional macroscale 3D printing techniques as enabled by commercially available machines, not microfabrication processes. In the context of microfabrication, we have added further references on the fabrication of non-polymeric materials to the revised manuscript, though those have the drawback that the trade-off between resolution and field of view still remains a challenge. In response to similar comments from Reviewer 3, we have also included the broader class of conventional metal and ceramic additive manufacturing in our statement. The sentence in question has been modified as follows: “The current state of the art in commercial macroscale 3D printing technology enables architectures with hundreds of unit cells per dimension [28]. These methods, which fall within the broad category of additive manufacturing (AM), have been widely exploited for polymeric materials, which unfortunately are not suitable for wave propagation because of material-intrinsic damping. Although exciting new directions in metal and ceramic AM have emerged [29, 30, 31], careful control of printing resolution and material properties still poses challenges. Microfabrication techniques enabled by multi-photon lithography, traditionally using polymeric materials at significantly higher spatial resolution [32], have overcome this limitation [33, 34]. Such techniques, while growing rapidly, are still nascent in their ability to achieve large sample sizes close to 100 mm while maintaining sub-micrometer spatial resolution [35, 36], lesser so when spatially-graded non-periodic architectures are involved.”*

I do not think the arguments added to the introduction here are accurate. Even commercial entities are now advertising “wafer scale” production through 3D printing, including the availability of silica resins [<https://www.nanoscribe.com/en/applications/wafer-scale-production-through-3d-direct-printing/>, <https://www.nanoscribe.com/en/products/gp-silica/>].

Response: We acknowledge the reviewer’s comment on existence of commercial solutions for “wafer scale” production. First, the statement in question (in the revised manuscript) was aimed at reinforcing the strength of silicon microfabrication techniques over other additive manufacturing techniques (in response to comments from this and one other reviewer); hence, it is not conflicting with the reviewer’s critique.

Second, the commercial solution cited by the reviewer is much more advanced technology than commonly available systems (in large-scale cleanroom facilities). We expect, based on our experience, that the accessibility of academic groups (or even industries) to such specialized systems is limited. Consequently, we believe that the commercialization of certain technologies does not necessarily imply that they are commonly available for all fundamental or applied research related to the topic. This is evidenced by the relatively limited studies on waveguides with such large scale separation in the literature, and the previously presented comments from reviews in the literature on the limited scalability of current fabrication routes. We, of course, do not claim that these technologies cannot be used for similar purposes; on the contrary, these may prove powerful for even more advanced wave guide designs – yet those require design approaches like the one presented here.

Reviewer 2: *Author: While the overall scaling argument is true in principle, we kindly disagree with the reviewer on its applicability to the technique proposed in the the cited reference. It is crucial to take into account the technique itself. The scaling from 384 to 100,000 unit cells (for the same unit cell size) is non-trivial – the sample sizes are limited by the range of the galvanometer mirrors used in their setup (300 × 300 μm). To increase this range, the galvanometer scanning system must be coupled to a scanning stage. Hence, a simple linear time scaling from 384 to 100,000 unit cells is an unreasonable assumption. Such “stitching” methods have been shown to introduce defects in printed samples [33]. As mentioned before, while attempts at making such scalable samples are improving fast, the technology is still in the nascent/development stage. Finally, we reiterate that this technique has been shown to work for polymeric samples only as yet.*

I assert that this is incorrect, see the reply to the prior statement.

Response: As mentioned in our earlier response, we acknowledge the existence of commercial technology cited by the reviewer but have seen almost no use of this technology in fundamental studies of multiscale wave guide design and development.

Furthermore, we stand by our (contested) argument in our first response letter, where we claim that the two-photon lithography method of Zhang et al., Nano Letters 24.8 (2024) does not reasonably/accurately scale to a build volume over 200 times larger than that presented in their original paper to achieve 100,000 unit cells. To back up our claim, we point to the following recent paper: Gu et al., *Nature*, 2025, <https://doi.org/10.1038/s41586-025-09842-x> (see their Figure 5), which thoroughly reviews micro/nanoscale additive manufacturing methods including Zhang et al.’s method, and also claims that it is restricted to a <1 mm² write field without error-prone stitching; this bottleneck has also been highlighted in a previous review by Saccone et al., *Nature*, 2022, <http://dx.doi.org/10.1038/s41586-022-05433-2>, referenced in our earlier response [36].

We acknowledge that micro-/nanoscale additive manufacturing is a rapidly advancing field, exemplified by the Gu et al. paper that we previously mentioned (which was recently published after our previous response), presenting advances in parallelized two-photon lithography. We believe such advances complement rather than conflict with the impact of our work, since we emphasize multiscale modeling and design advances that must parallel manufacturing advances to achieve technological progress.

Reviewer 3:

The authors have revised their manuscript in a thorough manner and addressed most reviewer comments in an adequate fashion. The argumentation in regards to sufficient novelty is mostly convincing and the authors have tempered and modified incorrect/inaccurate claims.

Response: Thank you for the positive feedback.

However, in several responses the authors purely point to changes/additions in the supplementary materials. They should carefully consider whether statements/claims are needed in the main manuscript, considering that supplementary material is only viewed by a fraction of readers. One example would be the computational cost and gains of the ray tracing approach, which is not reflected in the main manuscript.

Response: Good point. We agree that some of our previous revisions are worth discussing in the main text to ensure they reach the majority of readers. We have modified the *Modular Waveguide Design* section of the main text to include brief discussions about ray tracing validity, applicability of our optimization framework to other objectives, and computational efficiency of transient finite element simulations versus ray tracing (while honoring the strict word count limit imposed by the journal).

Reviewer 3: A note regarding the PDF manuscript, it is quite large, especially due to the figures. When viewing the manuscript in a browser, Fig. 4 does not load completely (it does load when viewing in Acrobat). In general, all figures load slowly and may even impede scrolling. The authors should make sure that figures (esp. pixel graphics) only use the resolution necessary for normal viewing and check their PDF in different viewers.

Response: We apologize for the excessive file size and have reduced the figure resolution.

References

- [1] Brillouin, L. Wave propagation in periodic structures. *McGraw-Hill* **2** (1946).
- [2] Hussein, M. I., Leamy, M. J. & Ruzzene, M. Dynamics of phononic materials and structures: Historical origins, recent progress, and future outlook. *Applied Mechanics Reviews* **66**, 040802 (2014).
- [3] Brûlé, S., Javelaud, E., Enoch, S. & Guenneau, S. Experiments on seismic metamaterials: molding surface waves. *Physical Review Letters* **112**, 133901 (2014).
- [4] Manushyna, D. *et al.* Application of vibroacoustic metamaterials for structural vibration reduction in space structures. *Mechanics Research Communications* **129**, 104090 (2023).
- [5] Dubček, T. *et al.* In-sensor passive speech classification with phononic metamaterials. *Advanced Functional Materials* **34**, 2311877 (2024).
- [6] Chen, Z., Guo, B., Yang, Y. & Cheng, C. Metamaterials-based enhanced energy harvesting: A review. *Physica B: Condensed Matter* **438**, 1–8 (2014).
- [7] Chaplain, G. J. *et al.* The 2024 acoustic metamaterials roadmap. *Journal of Physics D: Applied Physics* **58**, 433001 (2025). URL <https://doi.org/10.1088/1361-6463/add306>.
- [8] Jin, Y. *et al.* The 2024 phononic crystals roadmap. *Journal of Physics D: Applied Physics* **58**, 113001 (2025). URL <https://doi.org/10.1088/1361-6463/ad9ab2>.
- [9] Davies, B. *et al.* Roadmap on metamaterial theory, modelling and design. *Journal of Physics D: Applied Physics* **58**, 203002 (2025). URL <https://doi.org/10.1088/1361-6463/adc271>.
- [10] Ma, J., Xi, X. & Sun, X. Experimental demonstration of dual-band nano-electromechanical valley-hall topological metamaterials. *Advanced Materials* **33**, 2006521 (2021).

- [11] Bigoni, D., Guenneau, S., Movchan, A. B. & Brun, M. Elastic metamaterials with inertial locally resonant structures: Application to lensing and localization. *Physical Review B—Condensed Matter and Materials Physics* **87**, 174303 (2013).
- [12] Jin, Y., Djafari-Rouhani, B. & Torrent, D. Gradient index phononic crystals and metamaterials. *Nanophotonics* **8**, 685–701 (2019).
- [13] De Ponti, J. M. *et al.* *Graded elastic metamaterials for energy harvesting* (Springer, 2021).
- [14] Maspero, F. *et al.* Phononic graded meta-mems for elastic wave amplification and filtering. *Journal of Microelectromechanical Systems* (2023).
- [15] Tol, S., Degertekin, F. L. & Erturk, A. Phononic crystal Luneburg lens for omnidirectional elastic wave focusing and energy harvesting. *Applied Physics Letters* **111** (2017).
- [16] Zhu, J. *et al.* Acoustic rainbow trapping. *Scientific Reports* **3**, 1728 (2013).
- [17] Meng, H. *et al.* 3d rainbow phononic crystals for extended vibration attenuation bands. *Scientific Reports* **10** (2020). URL <http://dx.doi.org/10.1038/s41598-020-75977-8>.
- [18] Dorn, C. & Kochmann, D. M. Conformally graded metamaterials for elastic wave guidance. *Extreme Mechanics Letters* **65**, 102091 (2023).
- [19] Trainiti, G., Rimoli, J. J. & Ruzzene, M. Wave propagation in undulated structural lattices. *International Journal of Solids and Structures* **97**, 431–444 (2016).
- [20] Lin, S.-C. S., Huang, T. J., Sun, J.-H. & Wu, T.-T. Gradient-index phononic crystals. *Physical Review B—Condensed Matter and Materials Physics* **79**, 094302 (2009).
- [21] Chen, H. & Chan, C. T. Acoustic cloaking and transformation acoustics. *Journal of Physics D: Applied Physics* **43**, 113001 (2010).
- [22] Nassar, H., Chen, Y. & Huang, G. Polar metamaterials: a new outlook on resonance for cloaking applications. *Physical Review Letters* **124**, 084301 (2020).
- [23] Nolde, E., Craster, R. & Kaplunov, J. High frequency homogenization for structural mechanics. *Journal of the Mechanics and Physics of Solids* **59**, 651–671 (2011).
- [24] Ramirez, L. A. P., Erel-Demore, F., Rizzi, G., Voss, J. & Madeo, A. Effective surface forces and non-coherent interfaces within the reduced relaxed micromorphic modeling of finite-size mechanical metamaterials. *Journal of the Mechanics and Physics of Solids* **186**, 105558 (2024).
- [25] Demore, F., Rizzi, G., Collet, M., Neff, P. & Madeo, A. Unfolding engineering metamaterials design: Relaxed micromorphic modeling of large-scale acoustic meta-structures. *Journal of the Mechanics and Physics of Solids* **168**, 104995 (2022).
- [26] Dorn, C. & Kochmann, D. M. Ray theory for elastic wave propagation in graded metamaterials. *Journal of the Mechanics and Physics of Solids* **168**, 105049 (2022).
- [27] Dorn, C. & Kochmann, D. M. Inverse design of graded phononic materials via ray tracing. *Journal of Applied Physics* **134** (2023).
- [28] Telgen, B. *et al.* Rainbow trapping of out-of-plane mechanical waves in spatially variant beam lattices. *Journal of the Mechanics and Physics of Solids* **191**, 105762 (2024).
- [29] Schaeffer, M., Trainiti, G. & Ruzzene, M. Optical measurement of in-plane waves in mechanical metamaterials through digital image correlation. *Scientific Reports* **7** (2017). URL <http://dx.doi.org/10.1038/srep42437>.
- [30] Cha, J., Kim, K. W. & Daraio, C. Experimental realization of on-chip topological nanoelectromechanical metamaterials. *Nature* **564**, 229–233 (2018). URL <http://dx.doi.org/10.1038/s41586-018-0764-0>.

- [31] Shaikkea, A., Cui, H., O'Masta, M., Zheng, X. R. & Deshpande, V. S. The toughness of mechanical metamaterials. *Nature Materials* **21**, 297–304 (2022).
- [32] Wormser, M., Wein, F., Stingl, M. & Körner, C. Design and additive manufacturing of 3d phononic band gap structures based on gradient based optimization. *Materials* **10**, 1125 (2017). URL <http://dx.doi.org/10.3390/ma10101125>.
- [33] Kruisová, A. *et al.* Ultrasonic bandgaps in 3d-printed periodic ceramic microlattices. *Ultrasonics* **82**, 91–100 (2018). URL <http://dx.doi.org/10.1016/j.ultras.2017.07.017>.
- [34] Rice, H., Kennedy, J., Göransson, P., Dowling, L. & Trimble, D. Design of a kelvin cell acoustic metamaterial. *Journal of Sound and Vibration* **472**, 115167 (2020). URL <http://dx.doi.org/10.1016/j.jsv.2019.115167>.
- [35] Bauer, J. *et al.* Nanolattices: an emerging class of mechanical metamaterials. *Advanced Materials* **29**, 1701850 (2017).
- [36] Saccone, M. A., Gallivan, R. A., Narita, K., Yee, D. W. & Greer, J. R. Additive manufacturing of micro-architected metals via hydrogel infusion. *Nature* **612**, 685–690 (2022). URL <http://dx.doi.org/10.1038/s41586-022-05433-2>.
- [37] Kotz, F. *et al.* Two-photon polymerization of nanocomposites for the fabrication of transparent fused silica glass microstructures. *Advanced Materials* **33** (2021). URL <http://dx.doi.org/10.1002/adma.202006341>.
- [38] Skliutas, E. *et al.* Multiphoton 3d lithography. *Nature Reviews Methods Primers* **5** (2025). URL <http://dx.doi.org/10.1038/s43586-025-00386-y>.
- [39] Jonušauskas, L. *et al.* Mesoscale laser 3d printing. *Optics Express* **27**, 15205 (2019). URL <http://dx.doi.org/10.1364/OE.27.015205>.
- [40] Lee, J.-H., Singer, J. P. & Thomas, E. L. Micro-/nanostructured mechanical metamaterials. *Advanced Materials* **24**, 4782–4810 (2012).
- [41] Bückmann, T., Thiel, M., Kadic, M., Schittny, R. & Wegener, M. An elasto-mechanical unfeelability cloak made of pentamode metamaterials. *Nature Communications* **5**, 4130 (2014).
- [42] Meza, L. R. *et al.* Reexamining the mechanical property space of three-dimensional lattice architectures. *Acta Materialia* **140**, 424–432 (2017).
- [43] Harinarayana, V. & Shin, Y. Two-photon lithography for three-dimensional fabrication in micro/nanoscale regime: A comprehensive review. *Optics & Laser Technology* **142**, 107180 (2021).
- [44] Krödel, S. & Daraio, C. Microlattice metamaterials for tailoring ultrasonic transmission with elastoacoustic hybridization. *Physical Review Applied* **6**, 064005 (2016).
- [45] Kiefer, P. *et al.* A multi-photon (7 × 7)-focus 3d laser printer based on a 3d-printed diffractive optical element and a 3d-printed multi-lens array. *Light: Advanced Manufacturing* **4**, 28–41 (2024).
- [46] Zhang, L. *et al.* High-throughput two-photon 3d printing enabled by holographic multi-foci high-speed scanning. *Nano Letters* **24**, 2671–2679 (2024). URL <http://dx.doi.org/10.1021/acs.nanolett.4c00505>.
- [47] Li, Z. *et al.* Inverse design enables large-scale high-performance meta-optics reshaping virtual reality. *Nature Communications* **13**, 1–11 (2022).
- [48] Li, Z., Pestourie, R., Lin, Z., Johnson, S. G. & Capasso, F. Empowering metasurfaces with inverse design: principles and applications. *ACS Photonics* **9**, 2178–2192 (2022).
- [49] Krishnaswamy, S. *Photoacoustic Characterization of Materials*, 769–800 (Springer US, 2008). URL http://dx.doi.org/10.1007/978-0-387-30877-7_27.

- [50] Rogers, J. A., Maznev, A. A., Banet, M. J. & Nelson, K. A. Optical generation and characterization of acoustic waves in thin films: Fundamentals and applications. *Annual Review of Materials Science* **30**, 117–157 (2000). URL <http://dx.doi.org/10.1146/annurev.matsci.30.1.117>.
- [51] Vega-Flick, A. *et al.* Laser-induced transient grating setup with continuously tunable period. *Review of Scientific Instruments* **86** (2015). URL <http://dx.doi.org/10.1063/1.4936767>.
- [52] Moore, D. T. Gradient-index optics: a review. *Applied Optics* **19**, 1035–1038 (1980).
- [53] De Ponti, J. M. *et al.* Selective mode conversion and rainbow trapping via graded elastic waveguides. *Physical Review Applied* **16** (2021). URL <http://dx.doi.org/10.1103/PhysRevApplied.16.034028>.
- [54] Cervený, V. *Seismic ray theory* (Cambridge University Press Cambridge, 2001).
- [55] Born, M. & Wolf, E. *Principles of optics: Electromagnetic theory of propagation, interference and diffraction of light* (Elsevier, 2013).
- [56] Zelhofer, A. J. & Kochmann, D. M. On acoustic wave beaming in two-dimensional structural lattices. *International Journal of Solids and Structures* **115**, 248–269 (2017).
- [57] Kannan, V., Dorn, C., Drechsler, U. & Kochmann, D. M. Microscale architected materials for elastic wave guiding: Fabrication and dynamic characterization across length and time scales (2025). URL <https://arxiv.org/abs/2507.01757>. 2507.01757.
- [58] Sledzinska, M., Graczykowski, B., Alzina, F., Santiso Lopez, J. & Sotomayor Torres, C. Fabrication of phononic crystals on free-standing silicon membranes. *Microelectronic Engineering* **149**, 41–45 (2016). URL <http://dx.doi.org/10.1016/j.mee.2015.09.004>.
- [59] Olsson III, R. H., El-Kady, I. F., Su, M. F., Tuck, M. R. & Fleming, J. G. Microfabricated vhf acoustic crystals and waveguides. *Sensors and Actuators A: Physical* **145**, 87–93 (2008).
- [60] Hopkins, P. E. *et al.* Reduction in the thermal conductivity of single crystalline silicon by phononic crystal patterning. *Nano Letters* **11**, 107–112 (2010). URL <http://dx.doi.org/10.1021/nl102918q>.
- [61] Mohammadi, S., Eftekhari, A. A., Khelif, A., Hunt, W. D. & Adibi, A. Evidence of large high frequency complete phononic band gaps in silicon phononic crystal plates. *Applied Physics Letters* **92** (2008). URL <http://dx.doi.org/10.1063/1.2939097>.
- [62] Mustafazade, A. *et al.* A vibrating beam mems accelerometer for gravity and seismic measurements. *Scientific Reports* **10** (2020). URL <http://dx.doi.org/10.1038/s41598-020-67046-x>.
- [63] Jenni, L. V., Kumar, L. & Hierold, C. Hybrid lithography based fabrication of 3d patterns by deep reactive ion etching. *Microelectronic Engineering* **209**, 10–15 (2019). URL <http://dx.doi.org/10.1016/j.mee.2019.02.009>.
- [64] Loh, O., Vaziri, A. & Espinosa, H. D. The potential of MEMS for advancing experiments and modeling in cell mechanics. *Experimental Mechanics* **49**, 105–124 (2007). URL <http://dx.doi.org/10.1007/s11340-007-9099-8>.
- [65] Sharpe, W. N. *et al.* Strain measurements of silicon dioxide microspecimens by digital imaging processing. *Experimental Mechanics* **47**, 649–658 (2007). URL <http://dx.doi.org/10.1007/s11340-006-9010-z>.
- [66] Chasiotis, I. *Experimental Mechanics of MEMS and Thin Films*, 3–37 (Springer Netherlands, 2004). URL http://dx.doi.org/10.1007/978-94-007-1013-9_1.
- [67] Thelen, M., Bochud, N., Brinker, M., Prada, C. & Huber, P. Laser-excited elastic guided waves reveal the complex mechanics of nanoporous silicon. *Nature Communications* **12**, 3597 (2021).
- [68] Zega, V. *et al.* Microstructured phononic crystal isolates from ultrasonic mechanical vibrations. *Applied Sciences* **12**, 2499 (2022).

- [69] Hu, G., Tang, L., Liang, J., Lan, C. & Das, R. Acoustic-elastic metamaterials and phononic crystals for energy harvesting: A review. *Smart Materials and Structures* **30**, 085025 (2021).
- [70] Krushynska, A. O. *et al.* Emerging topics in nanophononics and elastic, acoustic, and mechanical metamaterials: an overview. *Nanophotonics* **12**, 659–686 (2023). URL <http://dx.doi.org/10.1515/nanoph-2022-0671>.
- [71] Charara, M., Kujala, Z., Lee, S. & Gonella, S. Spatially selective drop-motion programming using metamaterials. *Proceedings of the Royal Society A* **481**, 20240429 (2025).